# "Human displacements from tropical cyclone Idai attributable to climate change"

Benedikt Mester [1] [2], Thomas Vogt [1], Seth Bryant [2] [3], Christian Otto [1], Katja Frieler [1], and Jacob Schewe [1]

[1] Potsdam Institute for Climate Impact Research, Potsdam, Germany
[2] Institute of Environmental Science and Geography, University of Potsdam, Potsdam, Germany
[3] GFZ German Research Centre for Geosciences, Potsdam, Germany

Correspondence: Benedikt Mester (benedikt.mester@pik-potsdam.de)

# Abstract

Extreme weather events, such as tropical cyclones, often trigger population displacement. The frequency and intensity of tropical cyclones is affected by anthropogenic climate change. However, the effect of historical climate change on displacement risk has so far not been quantified. Here, we show how displacement can be partially attributed to climate change, using the example of the 2019 tropical cyclone Idai in Mozambique. We estimate the population exposed to high water levels following Idai's landfall, using a combination of a 2D hydrodynamical storm surge model and a flood depth estimation algorithm to determine inland flood depths from remote sensing images, for factual (climate change) and counterfactual (no climate change) mean sea level and maximum wind speed conditions. Our main estimates indicate that climate change has increased displacement risk from this event by approximately 12,600 - 14,900 additional displaced persons, corresponding to about 2.7 to 3.2%. The effect of wind speed intensification is larger than that of sea level rise. Besides highlighting the significant effects on humanitarian conditions already imparted by climate change, our study provides a blueprint for event-based displacement attribution.

# 1 Introduction

Between 1980 and 2021, an average of 45 tropical cyclones (TCs) globally have been recorded per year (Guha-Sapir et al., 2022). TCspose a set of societal risks to coastal communities around the world. While related monetary losses are high, with an average of US$ 57.2 billion every year since 2008 (Guha-Sapir et al., 2022), TCs also displace an average of 9.3 million people every year, with this hazard being responsible for 43% of all weather-related displacements (IDMC, 2022). Such forced displacements are associated with human suffering, as well as substantial financial costs (e.g., for providing shelter or from loss of economic production) and often require international assistance for disaster relief funds and humanitarian response (Desai et al., 2021).

At the same time, global climate change is expected to alter TC characteristics, resulting in an
increase in overall TC intensity (maximum wind speed and precipitation) and hence in the
frequency of very intense TCs (category 4-5 on the Saffir-Simpson scale), fundamentally
because of an increase in potential intensity due to warmer sea surface temperatures (SST)
(Emanuel, 2005, 2013, 1987; Knutson et al., 2020). Sea level rise (SLR), also driven by global
warming, additionally compound coastal flood risk associated with TCs (e.g., Garner Andra J.
et al., 2017; Lin et al., 2012; Resio and Irish, 2016). Historic TC data records are short and
partially inconsistent, making it difficult to determine the degree of intensification over time,
despite observed changes in some basins, such as the South Indian Ocean (Knutson et al.,
2019; Kossin et al., 2013, 2007; Webster et al., 2005). Moreover, existing TC datasets often
focus on maximum wind speed, neglecting coastal and inland flooding which may be the
dominant hazards, e.g., as for Hurricane Katrina or Hurricane Harvey (Bloemendaal et al.,
2021). Paleo climate records (Lin et al., 2014; Nott and Hayne, 2001) and synthetic TC tracks
(Bloemendaal et al., 2022, 2020; Emanuel et al., 2006) can be used to extend TC
records.However, sediment availability is limited to a few coastal stretches and the statistical
resampling process incorporates only the average observed climatic conditions, respectively,
hampering the assessment of global climate change impacts over longer time periods
(Bloemendaal et al., 2020). Nonetheless, given that global mean surface air temperature and
sea level have already risen above pre-industrial conditions by about 1.1°C and 0.20 m,
respectively (Gulev et al., 2021), it is likely that recent TC landfalls have caused more severe
societal impacts than would be expected without climate change. A probabilistic attribution
addressing this topic is limited by the shortness of TC records (Trenberth et al., 2015), and
may be additionally affected by multi-decadal variability (e.g., the Atlantic Multidecadal
Oscillation) or interannual climate variability (e.g., the El Niño–Southern Oscillation) (Patricola
and Wehner, 2018). As a consequence, the portion of TC-induced human displacements
attributable to climate change has so far not been quantified.
In this study, we address this research gap for the particular case of displacement triggered
by TC Idai in 2019. We examine the floods in central Mozambique associated with TC Idai,
considered to be "one of the Southern Hemisphere's most devastating storms on record"
(Warren, 2019). On the 14th of March, Idai made landfall near the densely populated port city
of Beira, inhabited by more than 530,000 people (Figure 1). Alongside strong winds (maximum
1-min sustained winds of 180 km/h) and extensive inland flooding caused by heavy rainfall,
the cyclone also created a storm surge of up to 4.4 m, leading to coastal flooding centered at
the port city of Beira (Probst and Annunziato, 2019). In Mozambique alone, TC Idai claimed
the lives of more than 600 people, and caused 478,000 internal displacements, as well as
widespread structural damage totaling more than US$ 2.1 billion (Guha-Sapir et al., 2022;
IDMC, 2022).
Here, we investigate how the coastal flooding would have manifested in a counterfactual world
without climate change, and consequently, how many of the observed human displacements
from TC Idai can be linked to climate change. For the attribution of the impacts we follow the
storyline approach introduced by Shepherd (Shepherd, 2016). To this end, we account for two
known mechanisms through which global climate change could have affected coastal flood
hazard: SLR and amplification of storm intensity. Storm track and size are not changed, even
though both parameters are subject to the effects of climate change (Knutson et al., 2020,
2019). We first estimate the influence of climate change on sea level and TC intensity in the
South Indian Ocean. We employ a high-resolution hydrodynamic flood model to simulate TC
Idai's peak coastal flood extent and depth, both under historical conditions and under
counterfactual conditions with lower sea levels and lower maximum wind speed,
corresponding to a world without climate change. We additionally use satellite imagery to
account for inland (fluvial and pluvial) flooding, and estimate the total number of people
affected by flooding. We then model the number of displacements based on flood depth-
specific vulnerability factors, and estimate the fraction of displacements that can be attributed
to climate change by comparing results under factual vs. counterfactual conditions.
We use an estimate of SLR that attempts to separate natural variability in ice sheet and glacier
mass balance and retain only the long-term trend induced by global warming (Strauss et al.,
2021). Beyond this, however, our analysis is indifferent to whether the trends in sea level and
TC intensity are anthropogenic or not. This is in line with the definition of *impact attribution* put
forward by the Intergovernmental Panel on Climate Change (IPCC), where "changes in
natural, human, or managed systems are attributed to [a] change in [a] climate-related system"
(O'Neill et al., 2022). Such a question can be separated from the *climate attribution* question
of whether the change in the climate-related system - here, sea level and TCs - is due to
anthropogenic forcing. This separation allows us to focus on the link between climate change
and displacement despite remaining uncertainty about the exact anthropogenic contribution.
We will return to this issue in the discussion.
This study aims to attribute coastal-flood induced human displacements from TC Idai to
historic climate change, using a quantitative modeling approach. It addresses the need for
insights on the human impacts of climate change globally, and in particular in countries like
Mozambique that suffer from a combination of high exposure to climate-related hazards - in
this case, TCs - and high socio-economic vulnerability. Moreover, Mozambique, like many
other countries, is characterized by limited availability of in-situ observational data and a lack
of calibrated, local-scale inundation models. We use remote-sensing data and a globally
applicable modeling framework to characterize flood exposure during TC Idai; reported
displacement data is retrieved from the Global Internal Displacement Database (GIDD). Our
approach is thus transferable to other cases in virtually all relevant countries.

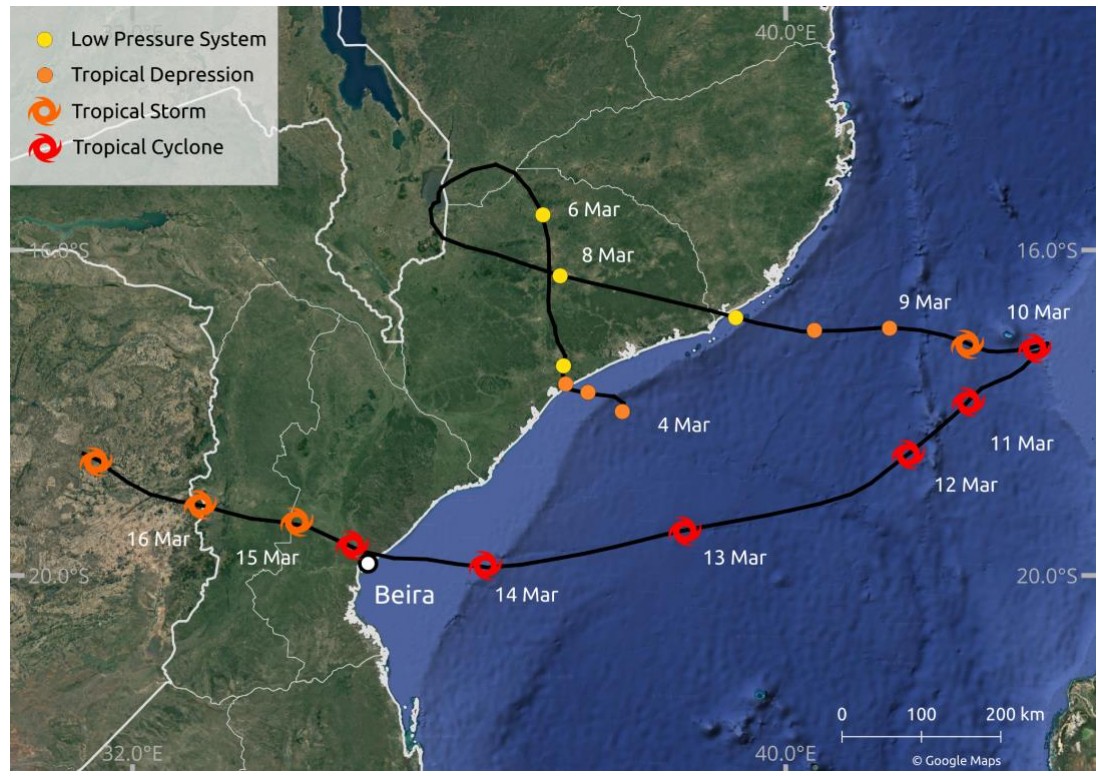

**Figure 1: Trajectory of tropical cyclone Idai over the South Indian Ocean.** Trajectory data is based on the IBTrACS database (Knapp et al., 2010). Mozambican administrative boundaries (GADM, 2018) in white; satellite image background by © Google Maps (Google Maps (a), 2022). Dates and tropical cyclone status adopted from ReliefWeb (ReliefWeb, 2019a).

# 2 Methods

## 2.1 Counterfactuals

Constructing counterfactuals for sea level and TC intensity requires estimating the effect of historical climate change on these quantities. Total global mean sea level has risen by approximately 23 cm since the turn of the 20th century (Church and White, 2011); at a rate that has increased over time (Dangendorf Sönke et al., 2017). According to the IPCC, it is very likely that the rate of global mean SLR was 1.5 (1.1 to 1.9) mm yr⁻¹ between 1902 and 2010, and 3.6 (3.1 to 4.1) mm yr⁻¹ between 2006 and 2015 (Gulev et al., 2021). Nonetheless, regional changes in sea level may differ substantially from the global average due to shifting surface winds, the differential expansion of warming ocean water, and the addition of melting ice, which can alter the ocean circulation (Fox-Kemper et al., 2021). Additionally, increases in the amount of water stored on land (due to construction of dams and reservoirs), as well as land subsidence, have also affected total sea level, with their relative effects varying geographically (Church et al., 2004; Strauss et al., 2021).

Long-term in-situ observational records of SLR are scarce in the Indian Ocean (Han et al., 2010), hampering a precise detection of changes in sea level. For example, no active tide

gauge stations can be found on the coast of Beira (Beal et al., 2019), with the nearest station located in Inhambane, Mozambique, 448 km south of Beira. However, regional historical SLR rates for Mozambique, derived from satellite imagery or models, are close to global mean estimates. IPCC rates of change in sea surface height (geocentric sea level) derived from satellite altimetry show regional SLR off the coast of Mozambique at around 4.0 mm yr⁻¹ for the period 1993–2012 (Church et al., 2013). Climate-induced SLR at the South-Eastern African coastline (1993 - 2015) is estimated at ~3.5 mm yr⁻¹ using a coastal-length weighted approach (Nicholls et al., 2021). Reconstructed sea level fields using global tide gauge data suggests global-averaged SLR at 1.8 ± 0.3 mm yr⁻¹ over the 1950-2000 period, with regional SLR off the coast of Mozambique at around 1.5 mm yr⁻¹ (Church et al., 2004). Han and colleagues (Han et al., 2010) estimate regional Mozambican SLR at approximately 1.2 mm yr⁻¹ between 1961-2008.

Given that these regional estimates are close to the global mean estimate by the IPCC, we assume that total SLR near Beira is the same as the global mean, a comparable approach as by Irish and colleagues (Irish et al., 2014). In order to exclude trends induced by natural variability, particularly in sea level contributions from glaciers and ice sheets, we use estimates of global mean sea level rise attributable to anthropogenic climate change for 1900–2012 from Strauss and colleagues (Strauss et al., 2021). Their ensemble estimate is 6.6 to 17.1 cm, which we use to define counterfactual sea level parameters for the coastal flood model. This also implies assuming no substantial local effects of land subsidence and human-induced changes in land water storage through reservoir construction and groundwater extraction that would confound comparison with the global estimates. This is hard to verify, but can be motivated by findings that city subsidence occurs only in a small fraction of the world's coasts (Nicholls et al., 2021).

Tropical cyclones are projected to become more intense with rising temperatures (Knutson et al., 2015), which is in line with the theoretical understanding of the potential intensity theory (Emanuel, 1987). Observed TC wind speed data in the South Indian Ocean basin shows that the maximum 10-minute sustained wind speed has been increasing by about 0.3 kn (0.15 m s⁻¹) per year on average, over the period 1973-2019 (Figure 2). Prior to 1973, the rate of increase was likely smaller, though observational data is lacking. We make a conservative assumption corresponding to 50 years of increase at a rate of 0.2 kn (0.1 m s⁻¹) per year, resulting in a total difference in maximum wind speed of approximately 10 kn (5.1 m s⁻¹). For the case of TC Idai with maximum observed 10-minute sustained wind speeds of 105 kn (54 m s⁻¹), this corresponds to a 10% reduction in maximum wind speed by removing climate change, which we adopt as a plausible assumption for counterfactual TC intensity.

This value is in line with the remote sensing-based estimates provided in Kossin et al. (2013), who find that lifetime maximum TC intensities in the SIO have increased by about 4.6 m/s over the period 1982-2009 (1.7 m/s per decade), which corresponds to 8.5% of TC Idai's maximum intensity. If this rate of increase is linearly extrapolated to 2019, it results in an increase of about 6.3 m/s (11.6%). Since the rate of increase has likely risen along with surface warming, and since our period of reference extends back to 1973 rather than 1982, a value of 12% might be a safer assumption for comparing the results of Kossin et al. (2013) with our own estimate.

To quantify the effect of uncertainty in the estimate of TC intensity change, we conduct two
sensitivity experiments, with counterfactual intensity lower than factual by 8.5% and 12%,
respectively, reflecting the SOI estimate of Kossin et al (2013) both directly and when
extrapolated for comparability with our own estimate.
We note that lower rates of change have been found in climate model-based studies. Knutson
et al. (2020) find a 6% increase in maximum intensity of SIO TCs per 2°C global mean surface
warming. When applied to the historical increase in global mean surface temperatures of
1.1°C, this would yield an increase of 3.3%. While these climate model estimates are important
both for assessing future changes and for understanding the underlying mechanisms of
observed trends, the remote-sensing based trend estimates are more relevant for informing
the construction of the counterfactual in our study.

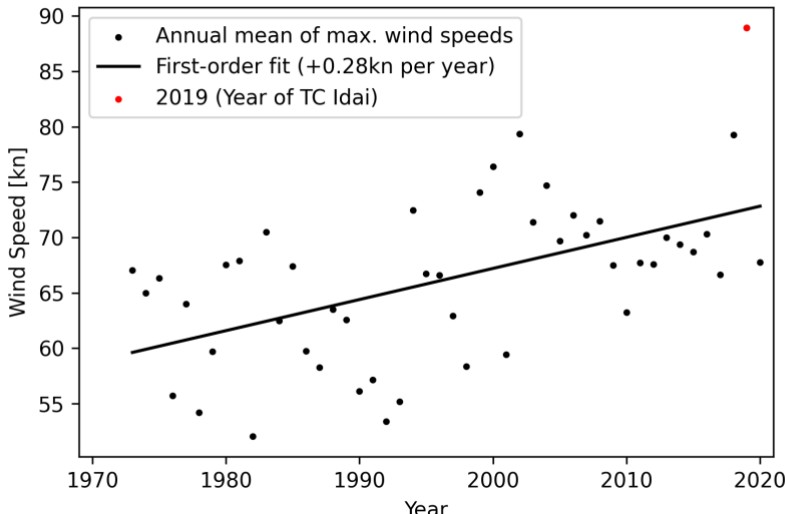


Figure 2: Annual means of maximum TC wind speeds in the South Indian Ocean (maximum
10-minute sustained wind speeds). Linear trend over the period 1973-2020; data from
IBTrACS database (Knapp et al., 2010).

## 214 2.2 Coastal Flood Modeling

The storm surge flood simulations are generated using the open-source geophysical flow
solver GeoClaw (Mandli and Dawson, 2014). GeoClaw uses an efficient adaptive mesh
refinement to model wind- and pressure-induced wave dynamics in the 2-dimensional depth-
averaged shallow water equations. The input data includes TC tracks, astronomical tides, and
topographical raster data (see below) and GeoClaw provides outputs in the form of gridded
maps of maximum flood heights as well as the temporal dynamics of storm surge at virtual
tide gauge locations. We configure GeoClaw to limit the automatic mesh refinement to a
spatial resolution of between 1 and 8 arc-seconds (approximately 30 and 240 m) inside of
Idai's landfall area and to between 100 and 900 arc-seconds (approximately 3 and 27 km) in
the open ocean.

As the factual input for GeoClaw, the TC track data from IBTrACS (Knapp et al., 2010)
provided by the WMO Regional Specialised Meteorological Center at La Reunion (operated
by MeteoFrance) is used. For the counterfactual scenarios with modified TC intensity, we
multiply all wind speed values along the track by a scalar factor of 0.9 (for a decrease of 10%
in intensity). The central pressure at each track position is increased by 0.1 times the
difference between central pressure and environmental pressure.
From the wind speed, pressure, and radius information provided along the TC track, GeoClaw
derives surface wind speeds and air pressure at arbitrary locations in space and time using a
radially symmetric wind profile (Holland, 1980) combined with the influence from the storm's
translational speed.
GeoClaw does not incorporate any tidal dynamics, nor meteorological forcings apart from the
TC wind and pressure fields mentioned above. To account for the influence of astronomical
tides, we configure GeoClaw to use an initial sea level according to gridded satellite altimetry
for 2019 (CMEMS, 2021), optionally enhanced by the minimum, mean, or maximum simulated
astronomical tides in the region of landfall according to the FES2014 global ocean tide atlas
(Lyard et al., 2021). For the counterfactual sea level scenarios, the amount of sea level rise
specified in the scenario description (between 6.5 and 17.0 cm) is subtracted from the initial
sea level.
The topographical input for GeoClaw is taken from digital elevation models (DEMs). We use
a combination of CoastalDEM 2.1 (Kulp and Strauss, 2021, 2018) in coastal areas, SRTM 15+
V2.3 (Tozer et al., 2019) over the open ocean and Multi-Error-Removed Improved-Terrain
(MERIT) DEM) (Yamazaki et al., 2019) everywhere else. All datasets are converted to the
same geoidal vertical datum (EGM96) at a spatial resolution of 9 arc-seconds (approximately
300 m). This resolution is the highest resolution where we were able to obtain numerically
stable results from GeoClaw. We note that no harmonization has been applied to make up for
disagreements between the different DEM products so that the transition from CoastalDEM
topography to SRTM 15+ bathymetry can be steep.
Due to a lack of tide gauges or suitable observed flood extent in Mozambique, it is not possible
to validate the performance of GeoClaw for TC Idai in the factual model runs. However, we
compare the water levels at a virtual tide gauge station off the coast of Beira, where the highest
impacts from TC Idai have been reported, with simulated water levels from the Global Tide
and Surge Model (GTSM) (Dullaart et al., 2021; Muis et al., 2020), and find the best agreement
of maximum surge heights for the GeoClaw run with the maximum astronomical tide
assumption, closely followed by the run assuming the monthly mean sea level (no tidal
adjustment) (Supplementary Figure S1).

## 265 2.3 Inland Flood Depth Estimation

Gridded depth maximums for the flood event  (Supplementary Figure S2) is calculated using
the Rolling HAND Inundation Corrected Depth Estimator (RICorDE) tool(Bryant et al., 2022)
supplied with terrain data from the MERIT DEM project, permanent surface water data from
the Joint Research Centre (JRC) Global Surface Water project (Pekel et al., 2016), and flood
extents from the FloodScan product (Atmospheric and Environmental Research & African Risk
Capacity, 2022). MERIT DEM provides a roughly 90 m resolution global layer derived from
multiple space-based sensors to minimize elevation errors. The maximum water extent layer
from JRC's Global Surface Water project provides a roughly 30 m resolution global layer of
locations detected as inundated on Landsat imagery (Wulder et al., 2016) from 1984-2019
(Pekel et al., 2016). Observed flood extents for TC Idai are obtained from Atmospheric and
Environmental Research & African Risk Capacity's accumulated 2-tier standard flood extent
depiction FloodScan product from 2019-03-01 to 2019-03-31 using the MERIT DEM
resolution. Originally developed for applications in Africa, this FloodScan algorithm relies on
satellite based low-resolution passive microwave data and was designed to capture national-
scale events. To accomplish this, the algorithm minimizes false-positives, making the
algorithm more prone to false-negatives and less sensitive to events with smaller spatial extent
and urban floods (Galantowicz and Picton, 2021). All data layers are re-projected to 90 m
resolution geodetic coordinates prior to the RICorDE computation.
RICorDE is a tool developed in pyQGIS for post-event analysis of fluvial flood events using
inundation masks derived from space-based observations. RICorDE first generates a Height
Above Nearest Drainage (HAND) grid followed by an inundation correction phase and a water
surface level (WSL) calculation phase. As part of pre-processing, the HAND grid is obtained
using WhiteboxTools' ElevationAboveStream (Lindsay, 2014) from the permanent surface
water layer and the DEM. In the first phase of RICorDE, the observed flood extents are
hydraulically corrected to account for under-predictions using the permanent surface water
layer and over-predictions using a HAND-derived inundation representing the upper quartile
of possible flooding extents. In the second phase, HAND values sampled from the inundation
shoreline are used to produce an interpolated WSL grid using WhiteboxTools' CostAllocation
algorithm (Lindsay, 2014). Finally, gridded water depths are obtained from this WSL grid
through subtraction with the DEM. RICorDE is explained in detail in the tool publication (Bryant
et al., 2022) and the source code can be accessed online
(https://github.com/cefect/RICorDE_pub).
The slower, more complex RICorDE algorithm has been shown to produce more accurate
depths maps for two fluvial flood events in Canada when compared to faster, more disaster
response-focused solutions like the Floodwater Depth Estimation Tool (FwDET) (Bryant et al.,
2022; Cohen et al., 2018). While no data is available to validate the performance of the depths
estimate for TC Idai, visual inspection suggests results are less accurate in areas with higher
elevation (>20 m), especially where drainageways are of comparable width to the resolution
of the JRC water extent layer. These false negatives in the JRC layer propagate as positive
bias in the HAND routine, which leads to higher elevation water surface predictions and similar
positive bias in the depth values (see white arrow in Figure S3a).

## 2.4 Combined Flood Depth Product

The inland flood depth estimates from RICorDE are resampled from 3 arcsec to 9 arcsec,
using the average resampling method (Rasterio library for Python), to match the resolution of
the GeoClaw output. All flood depths are rounded to the nearest decimeter, their outline is
cropped to the area of interest, and the final factual flood depth in each grid cell (shown in
Figure 3a) is determined as the maximum of both products. This accounts for both potentially

partly obscured satellite imagery by clouds and potential underestimation by the numerical model.

$$d_0 = max\,(\,d_{c,0}\,,d_r\,) \qquad (1)$$

with $d_0$ referring to the factual flood depth, and indices $c$ and $r$ referring to the coastal flood model (GeoClaw) and to the remote sensing data translated into flood depth using RICorDE, respectively. To derive the counterfactual flood depth $d_{cf}$, we subtract the difference between modeled factual and counterfactual coastal flood depths from the combined factual flood depth:

$$d_{cf} = d_0 - (\,d_{c,0} - d_{c,cf}\,) \qquad (2)$$

## 2.5 Displacement

We use displacement data from the publicly accessible GIDD, maintained by the *Internal Displacement Monitoring Centre* (IDMC, 2022). IDMC follows the definition of displacement provided in the Guiding Principles on Internal Displacement (OCHA, 2004), which states that "[i]nternally displaced persons are persons or groups of persons who have been forced or obliged to flee or to leave their homes or places of habitual residence, ... and who have not crossed an internationally recognized State border". This definition covers permanent displacement, temporary displacement, and pre-emptive evacuations (Gemenne, 2011), all summarized as "displacements" within our study. No granular information is available in GIDD on the type of displacement. Displacement numbers are based on multiple secondary sources, such as IOM, OCHA, or - in the case of TC Idai - the Mozambique National Institute of Disaster Management. The TC Idai event is categorized as a "storm" event, however, no information is given on how many of the displacements were caused respectively by flooding, strong winds, or a combination of both. Because of the extensive flooding observed in the wake of Idai's landfall and humanitarian reports often focused on flooding (ReliefWeb, 2019a), we assume in our main analysis that all displacements are caused by flooding (either coastal or inland). We assume that people exposed to flood levels greater or equal than 100 cm are affected by the flooding and thus prone to displacement, following previous studies (Custer and Nishijima, 2015; Kam et al., 2021). However, we also test the sensitivity of our results to this threshold choice by evaluating alternative water level thresholds of 10 cm and 50 cm. Our modeling approach assumes an artificially deterministic link between the TC hazard and displacement, which is adequate in the context of the factual-counterfactual approach where only one parameter - storm surge hazard - is modified while everything else, including vulnerability, is held constant. In general, the relationship between climatic events, pre-existing socio-economic conditions, and displacement is complex and only partially understood (Cattaneo et al., 2019; UK Government Office for Science, 2011). In other words, our study addresses the question of how many displacements might have occurred in a different climate but with the same vulnerability as observed; it does not address the question of how this vulnerability came about.

We first determine the flood extent with depths greater than the selected water level threshold and overlay it with population data to estimate the number of people affected. We use gridded

population data from GHS-POP (Schiavina et al., 2019) for the year 2015, on 9 arcsec resolution. Population growth in Mozambique was 1.12 % between 2015 and 2019 (The World Bank, 2022); we hence multiply all population grid cells with this factor, assuming a spatially equal population growth.

We then calculate the ratio between the number of observed displacements, and the number of affected people from the factual flood estimate. This ratio, which may be thought of as an event-specific displacement vulnerability factor, is different for every tide assumption, reflecting the uncertainty about the actual flood extent and depth. We compute for every impact level threshold $i$ and tide assumption $h$ a displacement vulnerability factor $v_{i,h}$ by dividing the number of observed displacements $D_o$ by the total number of affected people of the factual scenario $A_{i,h,o}$:

$$v_{i,h} = \frac{D_0}{A_{i,h,0}} \qquad (3)$$

Multiplying the specific displacement vulnerabilities with the counterfactual numbers of affected people, we derive the number of people at risk of displacement in a world without climate change. This means that the difference between factual and counterfactual displacement estimates comes only from differences in the flood hazard, while exposure and vulnerability factors are held fixed. We achieve this by multiplying $v_{i,t}$ with the number of affected people of the counterfactuals $A_{i,h,cf}$, and estimate the expected number of displacements for each counterfactual scenario $D_{i,h,cf}$:

$$D_{i,h,cf} = v_{i,h} * A_{i,h,cf} \quad (4)$$

We point out that the use of predefined flood thresholds implies the assumption that at a given flood depth, the risk of severe damages to, or even destruction of, residential buildings and other infrastructure typically becomes so large that people may be forced to flee. The number of people that actually become displaced then depends on additional physical, political and socio-economic factors, which may vary between local contexts and are not generally known. Their aggregate effect is reflected in the specific vulnerability factor vi,h. In other words, the link between flood hazard and displacement is "soft" in the sense that it is mediated by the local vulnerability. An alternative assumption would be that there is an (event-specific) flood-depth threshold below which there is no displacement, and above which people become displaced regardless; that is, a "hard" link between flood hazard and displacement. In this case, the flood-depth threshold could be derived directly from the data, as the depth level at which the calculated number of affected people equals the reported number of displacements. When we sum up the affected people per 10 cm flood depth increment for TC Idai, we obtain a threshold of about 400 cm (similar for all tide assumptions; Supplementary Table S1), for which the modeled number of affected people approximately equals the number of observed displacements. This value is very high in comparison with the thresholds cited further above, and we believe it is implausible for displacement to occur only in locations inundated by 4 meters or more. This exercise therefore lends further justification for the "soft link" approach.

Even though disaster reports for TC Idai suggest flooding to be the main driver of displacement, high wind speeds may have locally intensified the impact of TC Idai (Figure S4)

and be partially responsible for the observed displacements. We conduct an additional
analysis where we assume that people affected by either flooding or wind (or both) were at
risk of displacement with an equal vulnerability factor. We use a wind speed threshold of 96
kn (50 m s⁻¹) for population exposure (Geiger et al., 2018), corresponding to the Saffir–
Simpson scale classification 3 (major hurricane). The resulting wind field is overlaid with
gridded population data to compute the number of affected people, excluding those who are
already affected by flooding.

# 3 Results

## 3.1 Simulated flooding

We calculate storm surge flood extent and depth for the factual (driven with observed wind
speeds and sea levels) and counterfactual (reduced wind speeds and sea level) scenarios.
The difference between factual and counterfactual flooding (maximum tide, 10.5 cm SLR, 10%
TC intensification) is illustrated in the densely populated area of Beira (Figure 3b), the city
where TC Idai made landfall and destroyed 90% of all houses according to some disaster
reports (ReliefWeb, 2019b). Beira consists of two major population centers, of which the
southern one is close to the seaside and exhibits a higher population count.
Both factual and counterfactual flood extent covers the southern, highly populated part of Beira
(Figure 3c and 3d). The northern parts of the city are only marginally affected. Flood extents
are also similar between factual and counterfactual simulations in the areas east of Beira and
around the inflow of the Buzi River, located on the opposite side of the bay. Only a few isolated
locations no longer experience flooding after removing the effects of climate change.
In contrast, differences in simulated flood depth are more pronounced (Figure 3e).
Counterfactual flood depths are up to 80 cm lower than factual flood depth in some parts of
the southern city center. The highest difference in flood depth, of up to 140 cm, is found
between the northern and southern population centers of Beira. Flood depth differences
outside of Beira are rather low, however, Figure 3c and 3d show that absolute flood depths
drop below the critical flood depth of 100 cm over great parts around the west bank of the
Pungwe River inflow.

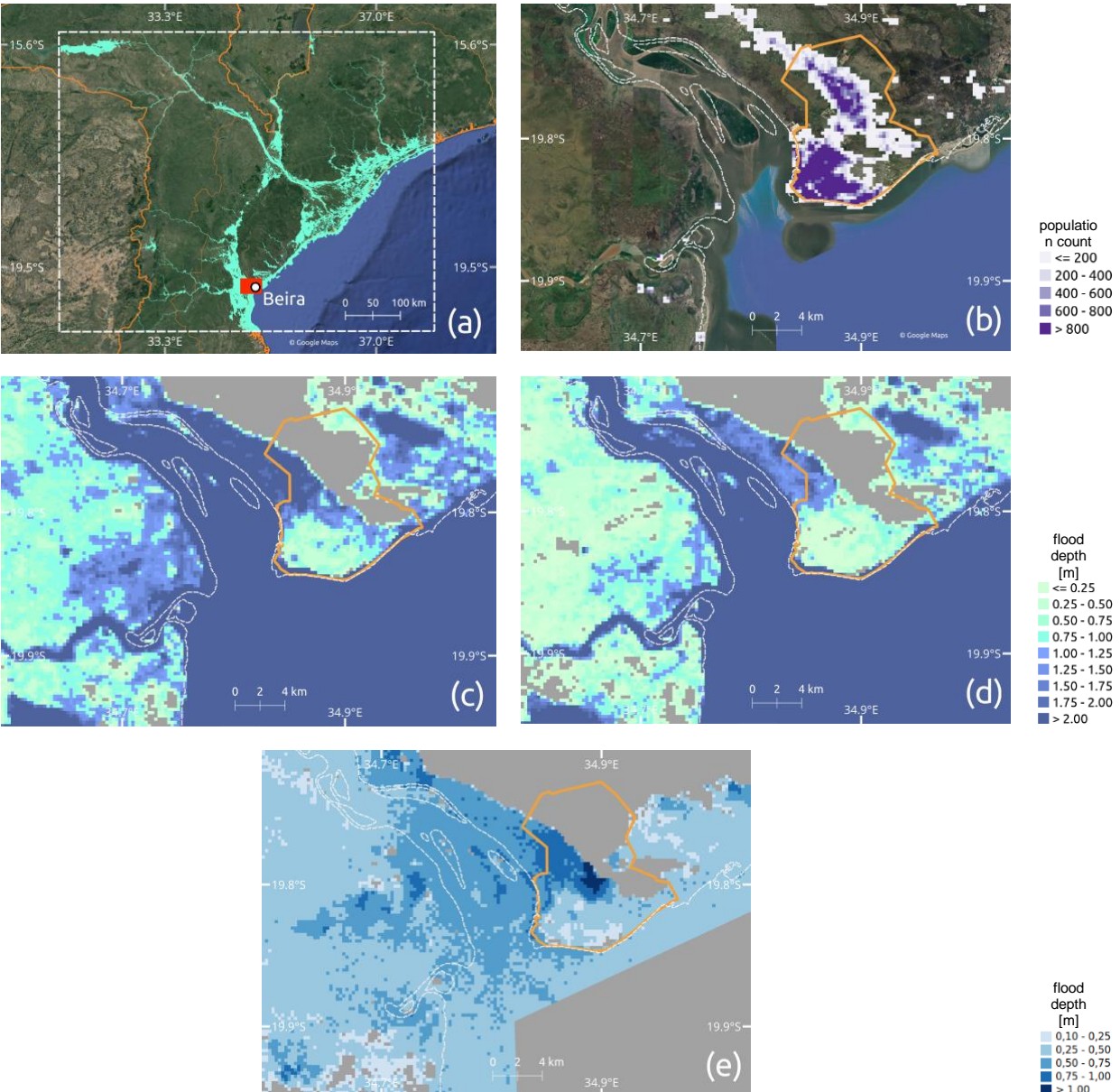


**Figure 3: Simulated flood extent for Mozambique; population distribution and inundation levels for the greater area of Beira.** (a) Combined factual estimate of inland and coastal flooding (binary; flood/no-flood). White d ashed box shows the area of interest in which flood exposure is computed. Red rectangle shows the extent of the section displayed in panel (b) - (e). (b) Population distribution for the greater area of Beira. Flood extent and levels for (c) the factual scenario (max. tide), and (d) the "counterfactual TC intensity + sea level rise (10.5 cm) - max. tide" scenario. Flood depth difference between (c) and (d) is displayed in (e). City neighborhoods of Beira (HDX, 2019) are indicated by orange lines and shoreline (Wessel and Smith, 1996) is represented by dashed white lines in (b) - (e); satellite image background by © Google Maps (Google Maps (b), 2022) in (a) and (b).


## 3.2 Displacement

In the next step, we investigate how the factual and counterfactual flood estimates translate into population at risk of displacement for the whole of Mozambique. We compare factual and

counterfactual affected people/displacements and compute the absolute relative change
based on the counterfactual results, representing the increase in impact due to climate
change. Our analysis shows that the intensification of TC wind speeds leads to an increase in
flood affected people and, consequently, in displacements by up to 2.7%, while
counterfactuals regarding the sea level lead to only small changes by up to 1.3 % (Figure 4,
Table 1 and Table S2). A combination of both counterfactuals only slightly exceeds the range
(increase by up to 3.2% for the maximum tide assumption) as in contrast when considering
the TC intensification alone. Despite the large uncertainty regarding SLR since 1900, the
difference in the number of people affected (or displaced) is rather marginal; being less than
1% increase between the largest and the smallest SLR estimate for the "cf SLR" simulations.
Our results highlight that the tide assumption plays a major role. The minimum and mean tide
lead to marginal changes in affected/displaced people, in contrast to the maximum
astronomical tide and monthly mean sea level from satellite altimetry (no tide), which show for
the "cf SLR + wind" simulations a median change in 3.0% (maximum change in 3.2%) and
2.7% (3.2%), respectively. Given the high number of affected people, already small changes
in the counterfactual scenarios lead to high changes in absolute numbers. The coupled effect
of higher wind speeds and higher sea level increases the number of affected people and
displacements by up to 39,300 and 14,900 (maximum tide) and 38,100 and 14,600 (monthly
mean), respectively. Results regarding impact flood levels of 10 cm and 50 cm are displayed
in Table 1 and the supplementary material (Figure S5 and S6), showing even higher changes
for the counterfactual scenarios of up to 56,500 displacements (13.4% increase).
Besides our central TC intensification assumption of 10%, we also examine two alternative
assumptions of 8.5% and 12% intensification, respectively, for the "max" tide (Figure 5). The
spread among the intensification scenarios is rather small, with median relative changes
varying between 2.9% and 3.7%. This translates to median estimates of 35,300 and 44,600
affected people, or 13,400 and 16,900 displacements, respectively (Table 1 and Table S2). In
contrast, the difference between the highest (4.0%) and lowest values (2.2%) is larger. In
absolute terms, this means a range of between approximately 27,400 and 48,200 affected
people, or 10,400 and 18,200 displacements.
We assume that high wind speed caused only a marginal fraction of displacements, following
disaster reports, media coverage and experience from other events; as an extreme example,
wind by Hurricane Sandy caused less than 0.01% of the overall damage (Strauss et al., 2021).
Nonetheless, in an additional sensitivity analysis, we also account for the number of people
affected by high TC wind speeds of 50 m s$^{-1}$ or above (Sect. Methods). Our analysis reveals
that the number of people affected not by flooding (maximum tide assumption, 100 cm impact
threshold) but by high wind speeds ranges between 340,900 to 360,600 in the factual
simulation. In the counterfactual, even the maximum wind speed attained in any grid cell
outside the flooded area drops from 51.5 m s$^{-1}$ to 46.3 m s$^{-1}$, i.e. below the above-mentioned
threshold; thus, no people are counted as affected. Assuming  the same vulnerability factor
for displacement due to high wind speed as due to flooding yields i 103,700 to 112,100
displacements, or 21.7 to 23.4% of the total displacement, attributable to climate change.

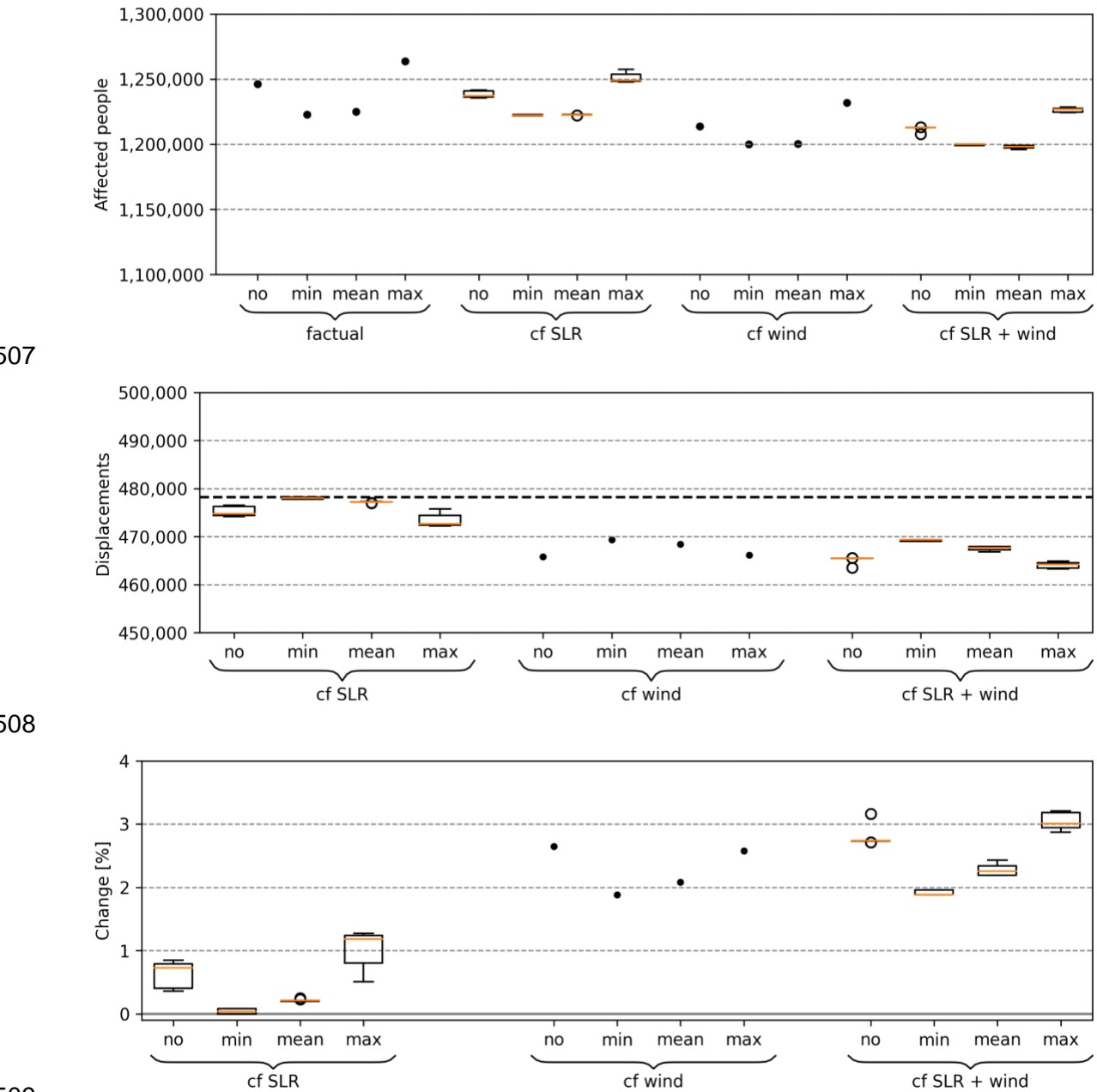




**Figure 4: Simulated affected people (top), displacements (middle) and percentage**
**change (bottom) for the 100 cm impact threshold.** The percentage change compares
factual and counterfactual displacements, and represents the absolute relative change based
on the counterfactual results. Three counterfactual scenarios are shown: lower sea level ("cf
SLR")), intensification ("cf wind"), and a combination of both ("cf SLR + wind"). Additionally, a
variety of counterfactual sea levels as well as a set of astronomical tides is presented, covering
minimum ("min"), mean ("mean"), and maximum ("max") as well as monthly mean sea level
from satellite altimetry ("no"). Bold dashed line in the middle panel shows the number of

observed displacements. Percentile changes in affected people and displacements are the same. The second quartile Q2 (median) of the box plot is shown in orange, "whiskers" are placed at ±1.5 * interquartile range (Q3-Q1).

Table 1: Overview main results for modeled displacement impact. Min./Median/Max. are related to the SLR scenarios. Orange background of the first results row indicates the primary parameter estimate. Cells with gray background indicate the altered parameter in comparison with the primary estimate.

| Counterfactual | Flood Depth Threshold [cm] | Intensification [%] | Tide | Displacements Dif. Min. | Displacements Dif. Median | Displacements Dif. Max | Displacements Dif. Min. [%] | Displacements Dif. Median [%] | Displacements Dif. Max [%] |
|---|---|---|---|---|---|---|---|---|---|
| SLR + wind | 100 | 10 | max | 13331 | 13958 | 14875 | 2.9 | 3.0 | 3.2 |
| SLR + wind | 100 | 10 | no | 12620 | 12740 | 14629 | 2.7 | 2.7 | 3.2 |
| SLR + wind | 100 | 10 | min | 8822 | 8822 | 9183 | 1.9 | 1.9 | 2.0 |
| SLR + wind | 100 | 10 | mean | 10235 | 10543 | 11353 | 2.2 | 2.3 | 2.4 |
| SLR + wind | 50 | 10 | max | 46695 | 49336 | 52275 | 10.8 | 11.5 | 12.3 |
| SLR + wind | 10 | 10 | max | 28557 | 32218 | 34456 | 6.4 | 7.2 | 7.8 |
| SLR | 100 | 10 | max | 2407 | 5584 | 5981 | 0.5 | 1.2 | 1.3 |
| wind | 100 | 10 | max | - | 12033 | - | - | 2.6 | - |
| SLR + wind | 100 | 8.5 | max | 10384 | 13354 | 14321 | 2.2 | 2.9 | 3.1 |
| SLR + wind | 100 | 12 | max | 14297 | 16870 | 18232 | 3.1 | 3.7 | 4.0 |

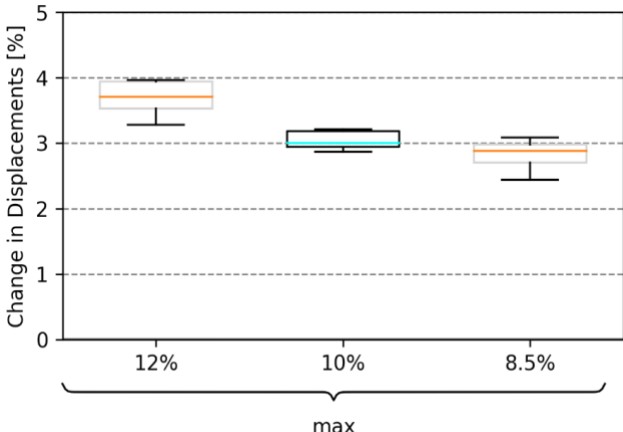

Figure 5: Percentage change in displacements between factual and counterfactual, for three different TC intensification assumptions. The percentage change compares factual and counterfactual displacements, and represents the absolute relative change based on the counterfactual results. The combined counterfactual scenario ("cf SLR + wind") with 100 cm impact threshold and the maximum astronomical tide ("max") is displayed. The central assumption of 10% intensification is highlighted with a cyan-colored median in the box plots. The second quartile Q2 (median) of the box plot is shown in orange/cyan, whiskers are placed at ±1.5 * interquartile range (Q3-Q1).

# 4 Discussion and conclusions

With more than one degree of global warming, most, if not all, extreme weather events now can be assumed to bear some imprint of climate change. By extension, this is also true for the humanitarian crises induced by catastrophic storms, floods, or droughts. However, while economic damages from climate change have been attributed both in case studies and global

studies (Frame et al., 2020b, 2020a; Sauer et al., 2021; Strauss et al., 2021), little is known
about the extent to which climate change has already exacerbated human displacement. Our
modeling study of TC Idai suggests that climate change may have induced between 12,600
(2.7%; lowest estimate under the no tide assumption) and 14,900 (3.2%; highest estimate
under the maximum tide assumption) additional displacements from this one event. This is
primarily due to the intensification of TC wind speed inducing a more powerful storm surge;
and to a lesser extent due to sea level rise providing a higher baseline for the storm surge.
We also show that the sensitivity of the results to the choice of TC intensification is
approximately in the same range as for the tide assumption.

Our results likely underestimate the full contribution of climate change to displacement
associated with TC Idai, because we solely addressed the effect of climate change on coastal
flooding, neglecting changes in inland flooding. Between March 3 and 17, heavy precipitation
between 200-400 mm was registered for Beira City and the region, with upstream sections of
the Pungwe River basin exposed to more than 600 mm (Probst and Annunziato, 2019). With
growing evidence that climate change not only affects precipitation intensity (Fowler et al.,
2021; Guerreiro et al., 2018; Scherrer et al., 2016) but also continental-scale changes in fluvial
flood discharge (Blöschl et al., 2019; Gudmundsson et al., 2021), it is likely that in a world
without climate change, the river flood magnitude would have been smaller, and even less
people would have been exposed than in our coastal-only counterfactual. Quantifying this
additional effect would require a river flood model capable of reproducing the observed flood
extent and associated inundation depths, and ideally  coupled with a coastal flood model to
capture the interaction between river flood and storm surge. Even though globally-applicable
frameworks for compound flood hazard modeling are under construction, and have recently
been tested for TC Idai (Eilander et al., 2022), evaluations of fluvial flood models reveal
important shortcomings in data-scarce regions such as Mozambique (Bernhofen et al., 2018;
Mester et al., 2021). Quantifying the role of river flooding in TC-induced displacement thus is
a timely challenge.

The inland river flood estimates based on satellite imagery exhibit several limitations and
uncertainties. In the absence of validation data, it is difficult to quantify the uncertainty arising
from the inland flood depths estimation. These gridded values are highly sensitive to the input
layers, namely the DEM (MERIT), permanent surface water (JRC), and the satellite-based
observation of inundation extents (FloodScan). Especially uncertainties regarding the choice
of DEM, used for both the inland flood depth estimation and the coastal flood model, should
not be neglected (Hawker et al., 2018). Qualitatively, the performance seems poor in areas
with higher elevations (>20m). This could be attributable to challenges in representing the
topography at 90 m resolution and dense obstructions that scatter returning signals (Shen et
al., 2019).

Similarly, no suitable validation data for the coastal flood simulations is available. According
to the AER description, the used products "depict large scale, inland river flooding well but are
less likely to depict flooding in smaller floodplains and near coastlines". We have hence opted
to not choose the AER product as the sole coastal flood hazard estimate nor as validation
dataset for the flood extent from our coastal flood model. A flood risk screening for Beira (van
Berchum et al., 2020) showed that simulated flood extent for a 10-year rainfall event plus a
10-year coastal surge event covers most parts of the Central and Munhava city districts of
Beira (South-Eastern city districts). In contrast, the satellite imagery by AER shows only little
flooding in this area, while it is assumed that flooding by TC Idai exceeded an average
recurrence interval of 10 years. For example, Emerton et al. (2020) show that GloFAS flood
forecasts indicated a 100% probability of exceeding the severe flood alert threshold (20-year
return period) for TC Idai at the Pungwe River (Emerton et al., 2020). Furthermore, newspaper
photographs (Bergensia, 2019) show flooding in the Area de Baixa part of Beira (Western
district of Beira), which was only partially flooded according to the satellite imagery. The AER
product thus likely underestimates flood extent, which  may be explained by cloud
obscurement or failure in automatic flood detection due to, for example, flooding in densely
populated areas, or the satellite passing over some time after the peak flooding when water
levels have already receded.
Furthermore, the coastal flood modeling framework does not incorporate any astronomical
tidal dynamics. Because there are no tide gauge records available in the region, we were only
able to compare the model's surge heights to the state-of-the-art Global Tide and Surge Model
(GTSM). For the derived flood maps, there were no observational benchmarks available for
validation. Moreover, the model is not able to take the interaction of the coastal surge with
increased river discharge at the estuaries into account. In some cases, this interaction has
been shown to influence water levels in a nonlinear way, for example for the 2016 Louisiana
flood (Bilskie and Hagen, 2018). Another source of uncertainty is again the DEM, in particular
the transition from topographic to bathymetric data at the coast lines.
Additionally, our analysis may be sensitive to the choice of population dataset (Archila Bustos
et al., 2020; Leyk et al., 2019), which may lead to uncertainties regarding our estimated
exposure. No information is available regarding the spatial distribution of displacements within
GIDD; we assume that vulnerability to displacement is uniform across the affected area. The
total number of displacements is furthermore not specifically categorized by hazard type,
which reflects the multivariate (wind, rain and flood) compound characteristic of TCs hazards
(Zscheischler et al., 2020). However, this impedes the attribution of coastal flood-induced
displacements. Furthermore, the GIDD estimates include different forms of displacement,
such as forced displacement or pre-emptive evacuations, with the latter potentially accounting
for a substantial proportion (McAdam, 2022). This poses far-reaching implications for
displacement risk modeling, as evacuations may already be triggered by lower flood depths,
or by early warnings of an impending hazard, which may not materialize in the expected
manner, or may not cause the level of destruction that would lead to a corresponding
magnitude of forced displacement.
Our main analysis also assumed no direct effect of high wind speeds on displacement, lacking
clear evidence for substantial displacement due to high winds alone. Our additional sensitivity
analysis suggests that changing this assumption could increase the number of displacements
attributable to climate change considerably. Given this potentially large effect, and our limited
understanding of the relative roles of different drivers of displacement in general, the specific
vulnerability to displacement from different types of hazard should be the subject of future
studies. Moreover, assuming that displacement can occur already at inundation depths of less
than 100 cm also leads to higher estimates of climate change-attributable displacement,
according to our sensitivity analysis. We also tested if the flood depth threshold can be
estimated from the data by summing up the affected people per 10 cm flood depth increment
until equaling the number of observed displacements. This analysis yields an alternative flood
depth threshold of 400 cm, which we assess to be physically not reasonable in the context of
building structure in Mozambique. Again, a better understanding of vulnerability beyond hard
physical flood depth thresholds and empirically derived vulnerability factors will be critical to
refine risk assessments. Future work may produce a functional relationship between
displacement risk, contextual drivers, and physical flood properties, covering, for example,
depth, velocity, and duration.
We did not change storm track or size in our counterfactual simulations. While storm tracks
may be affected by climate change (Knutson et al., 2019), we assume that Beira has not
become more or less likely as a landfall site. Mean storm size is found to increase
systematically with the relative sea surface temperature (Chavas et al., 2016), although
numerical simulations suggest that projected median sizes remain nearly constant globally
(Knutson et al., 2015). Assuming increases in storm size due to climate change would again
result in higher estimates of attributable displacements in our analysis.
By design, in our attribution study, we assumed a fixed population distribution in both factual
and counterfactual simulations, as well as a fixed, empirically determined displacement
vulnerability factor, and only investigated changes in displacement risk following from changes
in the physical characteristics of TC Idai and its impacts. Assessments of future risks - or of
past impacts - should not only take into account the intensification of physical hazards, but
also changes in exposure (Kam et al., 2021); as well as potential changes in vulnerability due
to social, economic, or technological developments. For instance, TC-related displacements
depend not only on the damage to housing, but also on other factors such as government
responsiveness or poverty levels (Cissé et al., 2022).
Here, we have chosen a storyline approach for the impact attribution instead of a more
traditional probabilistic attribution approach (Philip et al., 2020; Titley et al., 2016), as for
instance previously employed to attribute heavy precipitation of Hurricane Harvey
(Oldenborgh et al., 2017) to climate change. One reason is that for Mozambique neither the
complete time series of rainfall nor the high station density required by a probabilistic approach
(van Oldenborgh et al., 2021) are available. Reanalysis products for precipitation could be
used as an alternative, however, their quality depends on geographic location, so the use of
multiple reanalysis and/or observation products is recommended (Angélil et al., 2016).
Nonetheless, a climate attribution approach focusing on changes in the probability or intensity
of TCs in the South Indian Ocean due to anthropogenic forcing (O'Neill et al., 2022) could
guide the construction of counterfactual scenarios of the storyline approach. Further, in
contrast to the probabilistic approach, the storyline approach allows us to investigate the
driving factors involved, as well as their plausibility (Shepherd et al., 2018).
Framing the risk of tropical cyclones in the context of climate change in an event-specific rather
than a probabilistic manner also allows us to assign absolute numbers of attributable
displacements, which raises risk awareness in a more tangible way. The responsibility for
managing and reducing displacement risk lies primarily at the national and provincial level, but
often local authorities, organizations, and communities respond to displacement disasters
(Hollinger and Sienkevych, 2019). Demonstrating quantitatively how climate change affects
the societal risks associated with natural hazards may play an important role in raising
awareness, with different types of stakeholders, to the changing nature of such risks. It may
also incentivize governments to step up their efforts both in terms of planning and investing
into adaptation measures, and rapidly mitigating greenhouse gas emissions. The storyline
approach is particularly suited for highlighting the risk-amplifying effects of climate change in
a tangible and accessible way, based on a well-known event in the recent past (van den Hurk
et al., 2023). Estimates of the costs of displacement additionally highlight the adverse
economic aspects of climate change (Desai et al., 2021); average costs have been put at $310
per displaced person per year, though actual costs are heavily dependent on the country and
duration (days/weeks to years) (IDMC, 2019). Only 50.7% of the required Mozambique
Humanitarian Response Plan 2019 of US$m 620.5 was funded, demonstrating that climate
change poses an additional burden to insufficiently equipped financial aid resources.
Anticipating the intensification of tropical cyclones under future global warming (Knutson et
al., 2020) calls for enhancing adaptation measures as well as disaster relief and humanitarian
aid. The IPCC AR6 projects an additional global increase in mean sea level and surface
temperature of 0.44 m / 1.2°C (SSP1-2.6) and 0.77 m / 4.0°C (SSP5-8.5), relative to a baseline
of 1995-2014, by the end of the 21st century(Fox-Kemper et al., 2021; Lee et al., 2021). Even
though these increases may vary between basins, an enhanced displacement risk due to Idai-
like TCs needs to be accounted for in the next decades, especially if future changes in
exposure due to population growth and urbanization are considered.
Our study expands the scope of extreme event impact attribution to include displacement as
a societal impact dimension. In general, due to the lack of calibrated regional models and
gauge stations, only few attribution studies (Luu et al., 2021; Takayabu et al., 2015) focus on
storms - or any extreme weather events, for that matter - in low-income countries. This not
only limits our understanding of climate change effects on extreme events from a global
perspective, but also biases geographically the amount of knowledge and information
available to inform risk management and adaptation strategies (Otto et al., 2020). Our impact
attribution is built on global-scale datasets and models, which could be employed in other
relevant locations. Despite the discussed limitations and uncertainties inherent to this
approach, displacements could be similarly attributed to climate change for other major TCs
that occurred in data- and model-scarce regions, such as Typhoon Haiyan (Philippines; 4.1
million displacements) or Cyclone Amphan (India and Bangladesh; combined 4.95 million
displacements) (IDMC, 2022) The continuing increase in spatial resolution of global-scale
products will eventually allow for more granular displacement risk assessments, which
regional authorities could incorporate in urban development plans, zoning regulations or
required building codes (IDMC, 2019). Mozambique, like many countries, is exposed not only
to TCs but also other climate-related hazards, such as droughts, and at the same time facing
socio-economic challenges, making it all the more important to understand and anticipate risks
in a changing climate. Our approach may hence be extended to large-n impact attribution,
using, for example, global counterfactual climate datasets (Mengel et al., 2021).

# Code availability

The source code for this study is available from
https://github.com/BenediktMester/TC_Idai_attribution.

# Data availability

Satellite imagery is used with the permission of Atmospheric and Environmental Research & African Risk Capacity. Output of the flood depth algorithm, GeoClaw results, and TC Idai wind speed files can be accessed at https://zenodo.org/record/6907855 (Mester et al., 2022). GHS gridded population data is available at https://data.jrc.ec.europa.eu/dataset/jrc-ghsl-ghs_pop_gpw4_globe_r2015a#dataaccess.

National borders of Mozambique were obtained from https://gadm.org/data.html. For the trendline analysis of annual means of maximum wind speeds we use IBTraCS Version 4 database, accessible at https://www.ncei.noaa.gov/data/international-best-track-archive-for-climate-stewardship-ibtracs/v04r00/access/netcdf/IBTrACS.ALL.v04r00.nc.

All data used for the figures are publicly available. Maps were generated with QGIS, which can be downloaded at https://www.qgis.org/. Satellite imagery background by © Google Maps can be accessed via http://mt0.google.com/vt/lyrs=s&hl=en&x={x}&y={y}&z={z}. We used IBTrACS Version 4 to extract the trajectory data of tropical cyclone Idai, availabe at https://www.ncei.noaa.gov/products/international-best-track-archive?name=ib-v4-access. Mozambique admin level 4 shapefiles for Beira are available at https://data.humdata.org/dataset/mozambique-admin-level-4-beira-and-dondo-neighbourhood-boundaries. GSHHG shoreline data can be accessed via https://www.ngdc.noaa.gov/mgg/shorelines/data/gshhg/latest/.

# Author contributions

B.M. and J.S. designed the study, with contributions from T.V., C.O., and K.F. T.V. designed and performed coastal flood model calculations. S.B. estimated flood depths from satellite imagery. B.M. computed the number of affected people and displacements. B.M. and J.S. analyzed the results, and C.O. and K.F. contributed to the interpretation. B.M. and J.S. jointly wrote the paper, with contributions from T.V., S.B., and C.O.

# Competing interests

The authors declare no competing interests.

# Acknowledgments

This research received funding from the European Union's Horizon 2020 research and innovation programme under grant agreement No 820712 (RECEIPT).

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

# Supplementary Material

## "Human displacements from tropical cyclone Idai attributable to climate change"

Benedikt Mester [1] [2], Thomas Vogt [1], Seth Bryant [2] [3], Christian Otto [1], Katja Frieler [1], and Jacob Schewe [1]

[1] Potsdam Institute for Climate Impact Research, Potsdam, Germany
[2] Institute of Environmental Science and Geography, Potsdam University, Potsdam, Germany
[3] GFZ German Research Centre for Geosciences, Potsdam, Germany

Correspondence: Benedikt Mester (benedikt.mester@pik-potsdam.de)

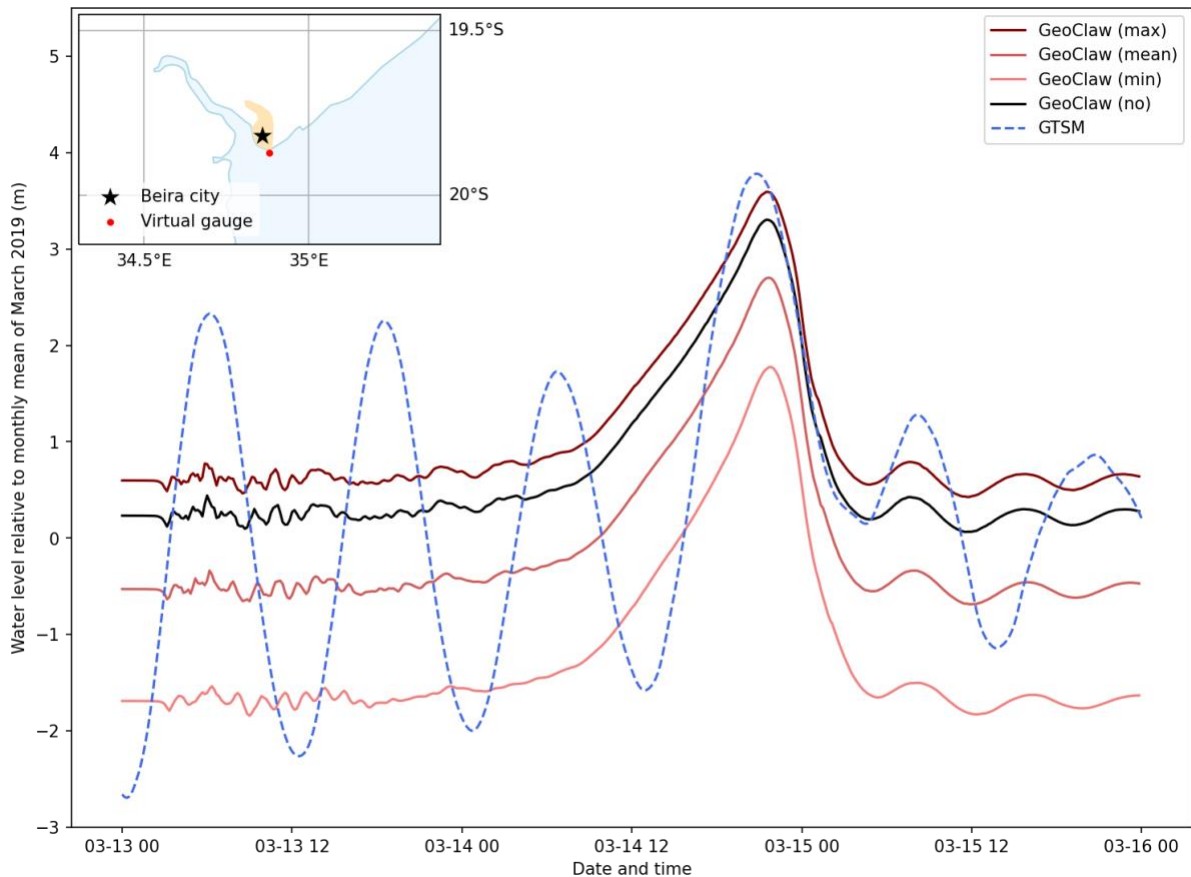

**Figure S1: Water levels at a virtual tide gauge station off the coast of Beira, Mozambique, according to simulations.** Several runs of GeoClaw are compared to GTSM: The GeoClaw runs are initialized with different base sea levels corresponding to assumptions of low (min), average (mean), and high (max) astronomical tides at landfall. Another run of GeoClaw is initialized with the monthly mean sea level from satellite altimetry (no). GTSM is driven by astronomical tidal forcing, and ERA5 meteorological forcing overlayed by a

parametric TC wind field. While GeoClaw does not incorporate tidal dynamics, the maximum
surge heights agree well with GTSM.

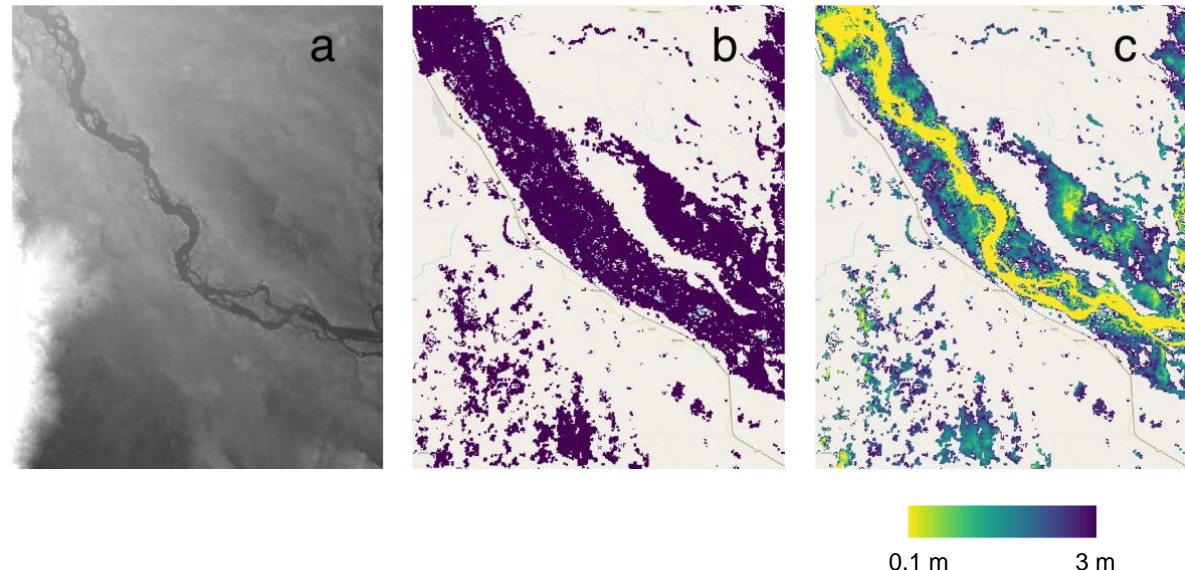

**Figure S2: Extract of a) MERIT DEM (Yamazaki et al., 2019), b) flood extents obtained**
**from FloodScan (Atmospheric and Environmental Research & African Risk Capacity**
**2022), and c) corresponding gridded-depths computed with the RICorDE algorithm.**
AFED-detected non-persistent water (2019/03/01 - 2019/03/31). Includes copyrighted material
of Atmospheric and Environmental Research, Inc. with its permission.

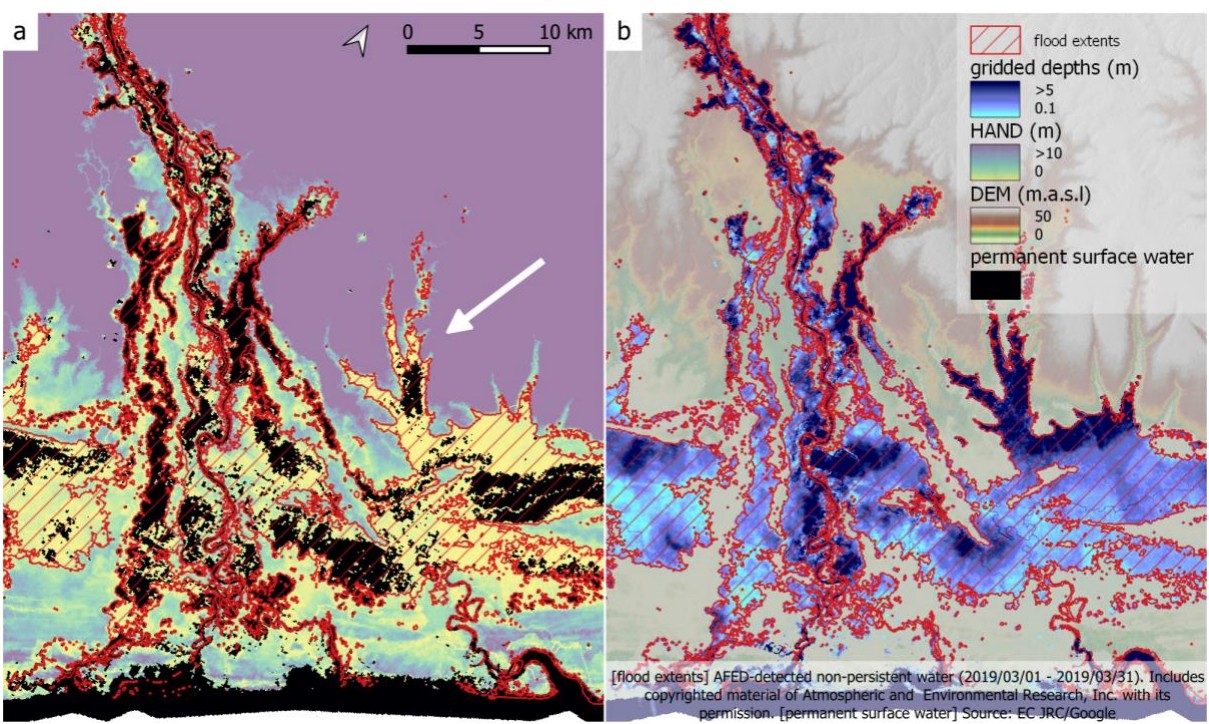


Table S1: Critical flood depths for which the simulated affected people approximately equals
the 478,000 reported displacements. The closest upper and lower 10 cm flood depth steps
are shown for each tide.

| Tide | Critical Flood Depth [cm] | Affected People |
|------|---------------------------|-----------------|
| no | 401-410 | 480,838 |
| no | 411-420 | 474,140 |
| max | 391-400 | 495,714 |
| max | 401-410 | 471,209 |
| min | 391-400 | 495,674 |
| min | 401-410 | 471,148 |
| mean | 391-400 | 502,572 |
| mean | 401-410 | 478,067 |


**Figure S3: Example of RICorDE performance against flood extents obtained from**
**FloodScan (Atmospheric and Environmental Research & African Risk Capacity 2022)**
**showing a) permanent surface water (Pekel et al., 2016) and resulting HAND values; and**
**b) MERIT DEM (Yamazaki et al., 2019) and resulting depths values.**

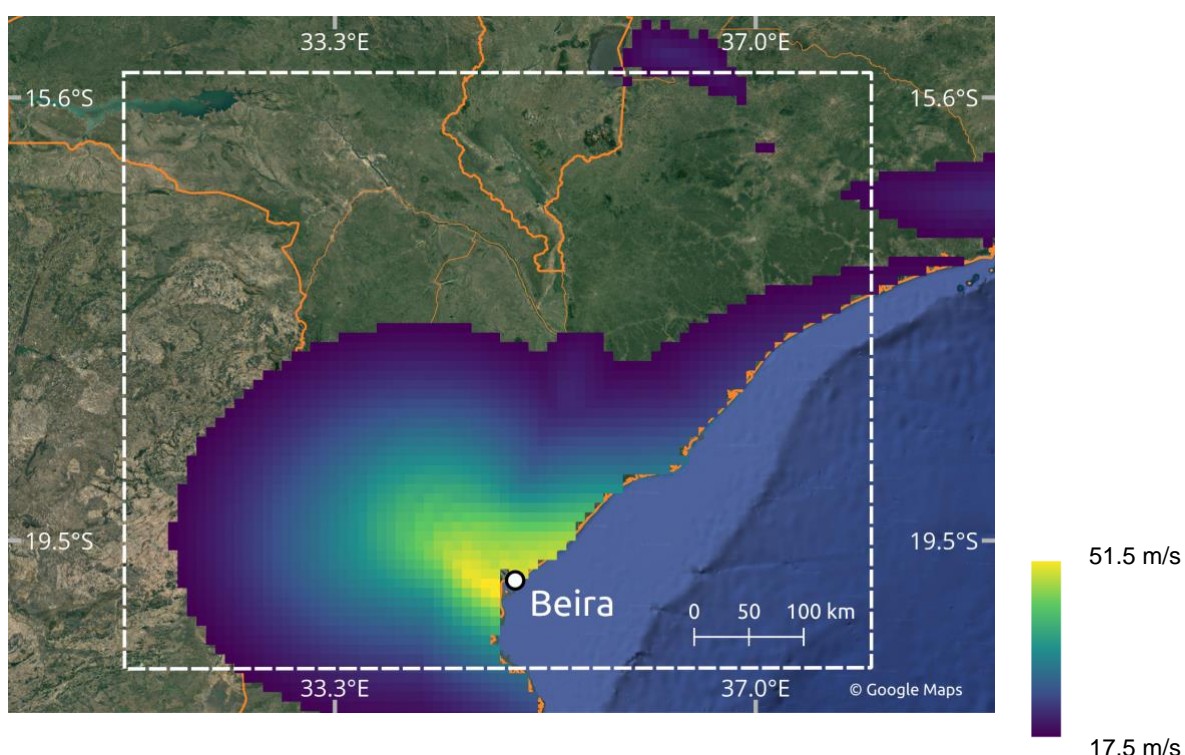

**Figure S4: Maximum wind speeds of TC Idai, which made landfall in Mozambique in**
**2019.** White dashed box shows the area of interest in which high wind speed exposure is
computed; satellite image background by © Google Maps (Google Maps (b), 2022).


Table S2: Overview main results for modeled affected people. Min./Median/Max. are related
to the SLR scenarios. Orange background of the first results row indicates the primary
parameter estimate. Cells with gray background indicate the altered parameter in comparison
with the primary estimate.

| Counterfactual | Flood Depth Threshold [cm] | Intensification [%] | Tide | Affected Dif.. Min. | Affected Dif.. Median | Affected Dif. Max | Affected Dif. Min. [%] | Affected Dif. Median [%] | Affected Dif. Max [%] |
|---|---|---|---|---|---|---|---|---|---|
| SLR + wind | 100 | 10 | max | 35229 | 36887 | 39311 | 2.9 | 3.0 | 3.2 |
| SLR + wind | 100 | 10 | no | 32886 | 33200 | 38121 | 2.7 | 2.7 | 3.2 |
| SLR + wind | 100 | 10 | min | 22557 | 22557 | 23481 | 1.9 | 1.9 | 2.0 |
| SLR + wind | 100 | 10 | mean | 26222 | 27012 | 29087 | 2.2 | 2.3 | 2.4 |
| SLR + wind | 50 | 10 | max | 161895 | 171054 | 181243 | 10.8 | 11.5 | 12.3 |
| SLR + wind | 10 | 10 | max | 123102 | 138884 | 148528 | 6.4 | 7.2 | 7.8 |
| SLR | 100 | 10 | max | 6360 | 14757 | 15805 | 0.5 | 1.2 | 1.3 |
| wind | 100 | 10 | max | - | 24934 | - | - | 2.6 | - |
| SLR + wind | 100 | 8.5 | max | 27443 | 35291 | 37847 | 2.2 | 2.9 | 3.1 |
| SLR + wind | 100 | 12 | max | 37784 | 44583 | 48181 | 3.1 | 3.7 | 4.0 |








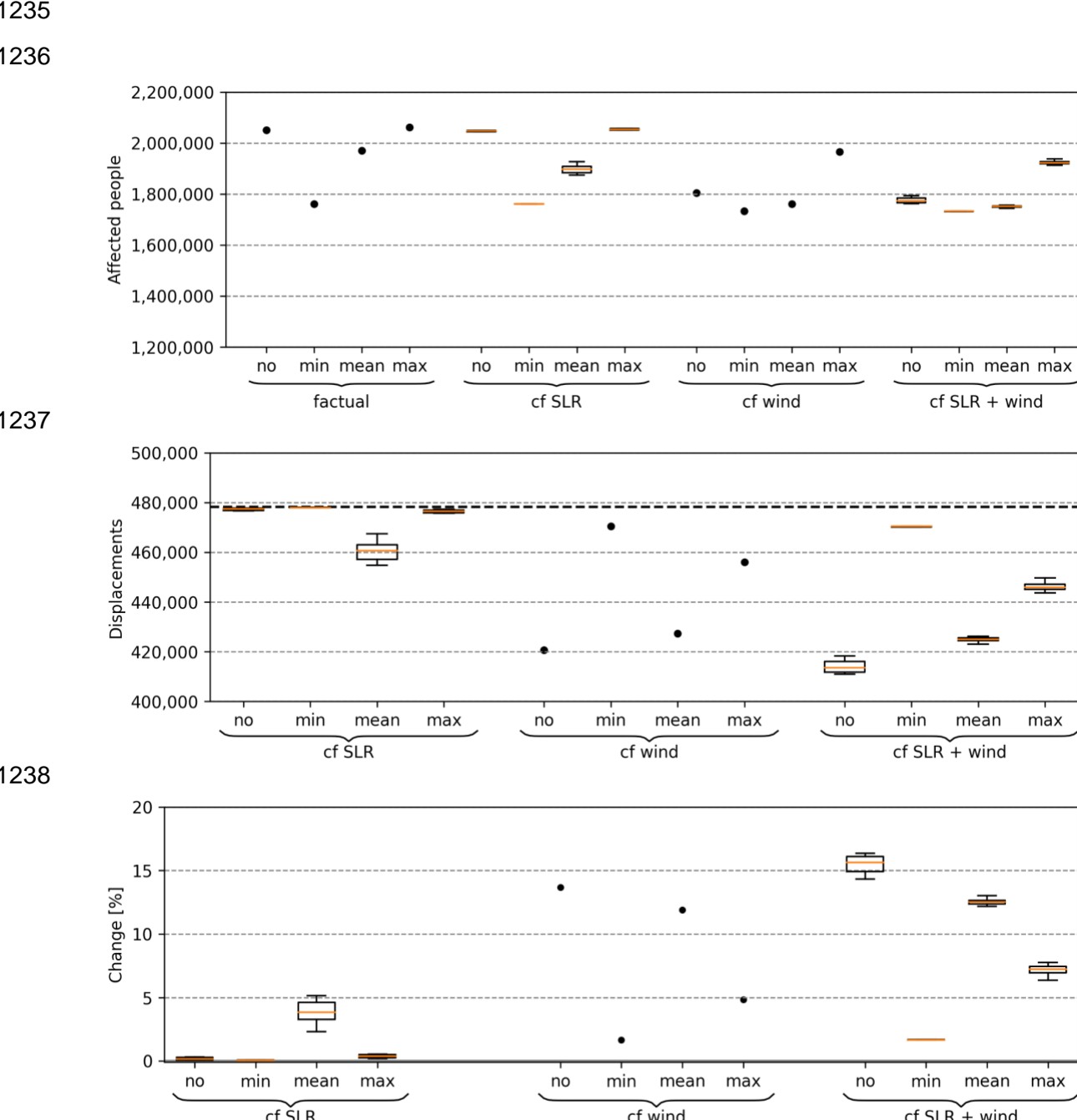



**Figure S5: Simulated affected people, displacements and percentige change by**
**flooding (10 cm impact threshold).** The percentage change compares factual and
counterfactual displacements, and represents the absolute relative change based on the
counterfactual results. Three counterfactual scenarios are shown: lower sea level ("cf SLR")),
intensification ("cf wind"), and a combination of both ("cf SLR + wind"). Additionally, a variety
of counterfactual sea levels as well as a set of astronomical tides is presented, covering
minimum ("min"), mean ("mean"), and maximum ("max") as well as monthly mean sea level
from satellite altimetry ("no"). Bold dashed line in the middle panel shows the number of
observed displacements. Percentile changes in affected people and displacements are the

same. The second quartile Q2 (median) of the box plot is shown in orange, "whiskers" are placed at ±1.5 * interquartile range ( Q3-Q1).

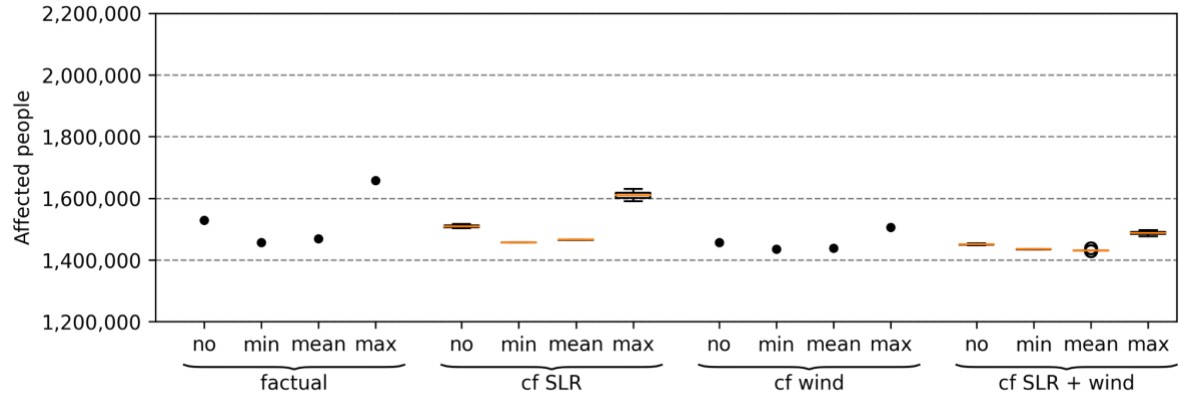

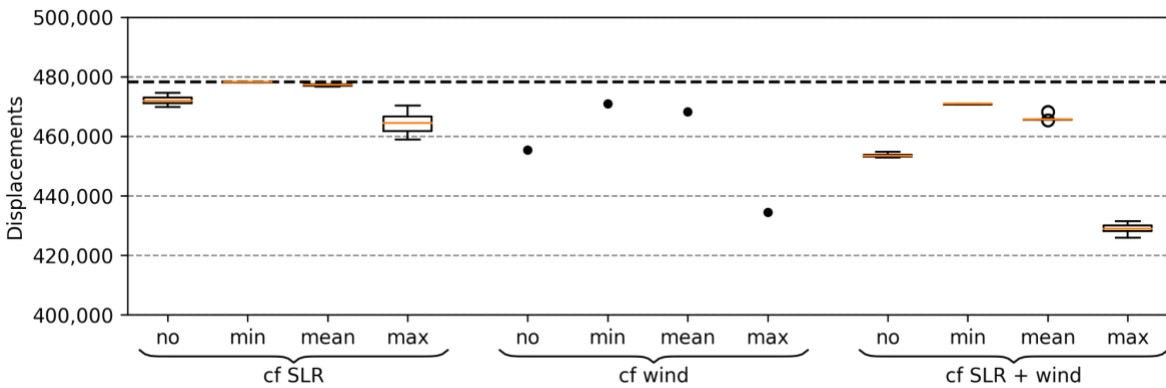

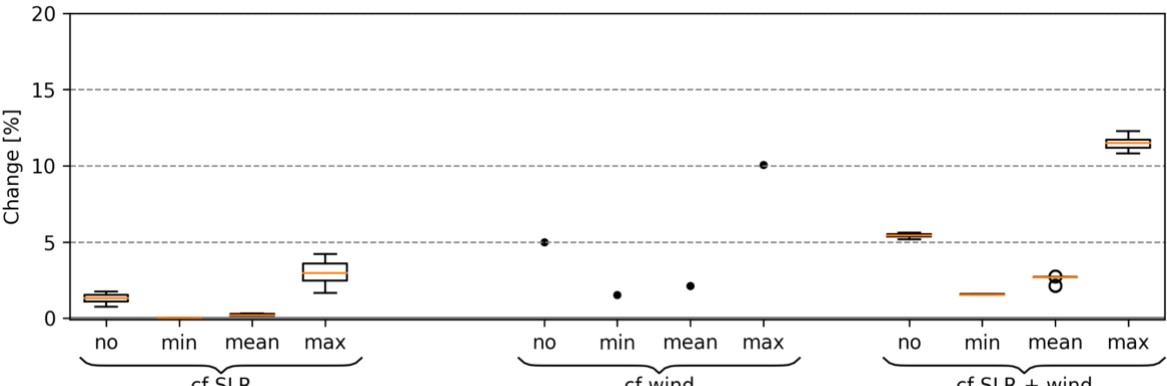

**Figure S6: Simulated affected people, displacements and percentige change by flooding (50 cm impact threshold).** The percentage change compares factual and counterfactual displacements, and represents the absolute relative change based on the counterfactual results. Three counterfactual scenarios are shown: lower sea level ("cf SLR")), intensification ("cf wind"), and a combination of both ("cf SLR + wind"). Additionally, a

variety of counterfactual sea levels as well as a set of astronomical tides is presented, covering minimum ("min"), mean ("mean"), and maximum ("max") as well as monthly mean sea level from satellite altimetry ("no"). Bold dashed line in the middle panel shows the number of observed displacements. Percentile changes in affected people and displacements are the same. The second quartile Q2 (median) of the box plot is shown in orange, "whiskers" are placed at ±1.5 * interquartile range (Q3-Q1).