# Peer review of "“Human displacements from tropical cyclone Idai attributable"

_EGUsphere, 2022_

## Author Comment (AC1)

We would like to thank both reviewers for their insightful suggestions and comments on the manuscript. Following both reviews, we have revised the manuscript in several ways and think that the changes have improved the quality in terms of readability, structure, and insights substantially. The main adjustments can be summarized as:

- Restructured and more in depth results section
- Sensitivity analysis of additional intensification rates of TC wind speeds
- Elaborated discussion on the nature of GIDD displacement data and the overall connectivity between forced human mobility, environmental and socio-economic contextual drivers.
- In depth discussion on the limitation and uncertainties regarding the coastal flood simulations, inland flooding estimations, population data, and used critical flood depth thresholds

In our opinion, the incorporation of these adjustments have improved the manuscript substantially thanks to your suggestions and comments. In the following, we respond (blue font) more in detail to every single referee comment (black font). Line references are related to the revised manuscript unless otherwise stated.

In addition to the responses to the review comments, we realized that a software issue in one of the dependencies of GeoClaw had been identified and corrected in the meantime. We have therefore updated the model code and repeated all experiments. This has led to moderate changes in our numerical results, resulting in somewhat lower estimates of the climate change impact on displacement than before (see also our response to reviewer #2, further below).

Review #1

RC1: 'Comment on egusphere-2022-1308', Anonymous Referee #1

The manuscript "Human displacements from tropical cyclone Idai attributable to climate change" presents a modeling study that assesses the impacts of observed climate change on population displacement risk of tropical cyclone (TC) Idai in Mozambique, using a storyline approach. The work presents an interesting and innovative approach to assessing and attributing TC-related risks, using displacements as the response variable. However, the manuscript in its current form has a number of major limitations which inhibit drawing clear take-home messages and conclusions from it. The following issues need to be addressed to qualify for publication:

Abstract

- The abstract is rather short, focusing mainly on the methods used in the study rather than on its results. To make it more concrete, more detail on the results of the study would be useful.

We have added the following main result to the Abstract:

"The effect of wind speed intensification is larger than that of sea level rise."

Introduction

- The introduction section effectively describes the relevance of the work, but uses vague language such as "immense" (line 29); "extensive human suffering" (line 35); "intense storm surge" (line 58). Citing numbers from the relevant literature would considerably improve the introduction.

We generally agree, even though we believe that some qualitative language in the Introduction is acceptable in order to convey general messages without distraction from too many numbers. We have removed some vague and unnecessary expressions, and have supplemented some other expressions with more concise terminology and numbers. For example, we have changed lines 99 to 102 of the Introduction:

"Alongside strong winds (maximum 1-min sustained winds of 180 km/h) and extensive inland flooding caused by heavy rainfall, the cyclone also created a storm surge of up to 4.4 m, leading to coastal flooding centered at the port city of Beira (Probst and Annunziato, 2019)."

- In total, the literature review provided in this section is rather limited and could e.g. include the work of Bloemendaal et al. 2020; 2021; 2022.

Thank you for this comment and the literature suggestions. We have extended the Introduction by giving an overview on the challenges related to the assessment of intensity changes of TCs due to climate change (lines 74-93):

"Historic TCs data records are short and partially inconsistent, making it difficult to determine the degree of intensification over time, despite observed changes in some basins, such as the South Indian Ocean (Knutson et al., 2019; Kossin et al., 2013, 2007; Webster et al., 2005). Moreover, existing TC datasets often focus on maximum wind speed, neglecting coastal and inland flooding which may be the dominant hazards, e.g., as for Hurricane Katrina or Hurricane Harvey (Bloemendaal et al., 2021). Paleo climate records (Lin et al., 2014; Nott and Hayne, 2001) and synthetic TC tracks (Bloemendaal et al., 2022, 2020; Emanuel et al., 2006) can be used to extend TC records. However, sediment availability is limited to a few coastal stretches and the statistical resampling process incorporates only the average observed climatic conditions, respectively, hampering the assessment of global climate change impacts over longer time periods (Bloemendaal et al., 2020). Nonetheless, given that global mean surface air temperature and sea level have already risen above pre-industrial conditions (by about 1.1°C and 0.20 m, respectively (Gulev et al., 2021)), it is likely that recent TC landfalls have caused more severe societal impacts than would be expected without climate change. A probabilistic attribution addressing this topic is limited by the shortness of TC records (Trenberth et al., 2015), and may be additionally affected by multi-decadal variability (e.g., the Atlantic Multidecadal Oscillation) or interannual climate variability (e.g., the El Niño–Southern

Oscillation) (Patricola and Wehner, 2018). As a consequence, the portion of TC-induced human displacements attributable to climate change has so far not been quantified."

- It is unclear to me whether "inland (freshwater) flooding" (line 73) refers to pluvial or fluvial flooding (or both?).

The satellite imagery of inland flooding captures flood extent of both types of flooding, we hence appended the sentence with this information:

"inland (fluvial and pluvial) flooding"

- The last sentence of the introduction states that the issues described in that paragraph (lines 79-89) will be picked up later again; however, I could not find the respective section.

Thank you for pointing this out. We have added the underlined sentence for clarification in the Discussion (lines 996-998):

"Here, we have chosen a storyline approach for the impact attribution instead of a more traditional probabilistic attribution approach … Reanalysis products for precipitation could be used as an alternative, however, their quality depends on geographic location, so the use of multiple reanalysis and/or observation products is recommended (Angélil et al., 2016). Nonetheless, a climate attribution approach focusing on changes in the probability or intensity of TCs in the South Indian Ocean due to anthropogenic forcing (O'Neill et al., 2022) could guide the construction of counterfactual scenarios of the storyline approach."

- The end of the introduction would profit from a clear statement of the study aims and how they are achieved in the remainder of the paper. Clarification of the fact that this is a modeling study that uses observed displacements as reported in the GIDD as its only empirical evidence would be useful as well.

Thank you for highlighting this. We have added the following paragraph at the end of the introduction. A more detailed discussion on the displacement data is given in "2.5 Displacement".

"This study aims to attribute coastal-flood induced human displacements from TC Idai to historic climate change, using a quantitative modeling approach. It addresses the need for insights on the human impacts of climate change globally, and in particular in countries like Mozambique that suffer from a combination of high exposure to climate-related hazards - in this case, TCs - and high socio-economic vulnerability. Moreover, Mozambique, like many other countries, is characterized by limited availability of in-situ observational data and a lack of calibrated, local-scale inundation models. We use remote-sensing data and a globally applicable modeling framework to characterize flood exposure during TC Idai; reported displacement data is

retrieved from the Global Internal Displacement Database (GIDD). Our approach is thus transferable to other cases in virtually all relevant countries."

Methods

- The methods, particularly 2.1 and 2.2, provide rather short and technical descriptions of the modeling work. I would suggest including more detail and reasoning behind the modeling choices and assumptions made, supported by additional literature. Some of these details can be found in the results section (section 3.1), which I suggest moving into the methods section as methodological choices are explained here. More specifically, I was wondering about the following points (not exhaustive):

We agree with these comments. We have moved section "3.1. Counterfactuals" in the beginning of the "Methods" section. This now illustrates to the reader our motivation for the modeling setup and the choices for the counterfactual runs, related to the next two questions.

- Lines 108-109: Why 10 % decrease?

The value of 10% is based on our own estimate of the trend in Southern Indian Ocean (SIO) tropical cyclone maximum wind speeds, using IBTracs data. It is, however, very much in line with the remote sensing-based estimates provided in Kossin et al. (2013), who find that lifetime maximum TC intensities in the SIO have increased by about 4.6 m/s over the period 1982-2009 (1.7 m/s per decade), which corresponds to 8.5% of TC Idai's maximum intensity. If this rate of increase is linearly extrapolated to 2019, it results in an increase of about 6.3 m/s (11.6%). Since the rate of increase has likely risen along with surface warming, and since our period of reference extends back to 1973 rather than 1982, a value of 12% might be a safer assumption for comparing the results of Kossin et al. (2013) with our own estimate. Thus, our choice of 10% is actually lower than the extrapolation of these published results, which are also extensively cited in the overview paper by Knutson et al. (2019).

In contrast, the Knutson et al. (2020) study is based on climate models rather than remote observations. The value of 5% per 2°C global mean surface warming refers to the median global estimate in that study; for the SIO, we extract a value of about 6% per 2°C from their results. When applied to the historical increase in global mean surface temperatures of 1.1°C, this would yield an increase of 3.3%. While these climate model estimates are important both for assessing future changes and for understanding the underlying mechanisms of observed trends, we believe the remote-sensing based trend estimates are more relevant for informing the construction of the counterfactual in our study. While the analysis in Kossin et al. (2013) is based on only about 30 years of data, our own analysis of almost 50 years worth of data (1973-2019) leads to a very similar result.

We have moved the first part of the Results section into the Methods section, such that the construction of the counterfactuals is explained first, right before the explanation of the coastal flood modeling. Section 3.1. now lays out this analysis (lines 225-233):

"Observed TC wind speed data in the South Indian Ocean basin shows that the maximum 10-minute sustained wind speed has been increasing by about 0.3 kn (0.15 m s⁻¹) per year on average, over the period 1973-2019 (Figure 2). Prior to 1973, the rate of increase was likely smaller, though observational data is lacking. We make a conservative assumption corresponding to 50 years of increase at a rate of 0.2 kn (0.1 m s⁻¹) per year, resulting in a total difference in maximum wind speed of approximately 10 kn (5.1 m s⁻¹). For the case of TC Idai with maximum observed 10-minute sustained wind speeds of 105 kn (54 m s⁻¹), this corresponds to a 10% reduction in maximum wind speed by removing climate change, which we adopt as a plausible assumption for a counterfactual TC intensity."

To explore the effects of the undeniable uncertainties in these estimates, we have added to our central run with 10% TC intensity increase, two runs assuming 8.5% and 12% increase respectively, reflecting the SOI estimate of Kossin et al (2013) both directly and when extrapolated for comparability with our own estimate:

Methods (lines 235-259):

"This value is in line with the remote sensing-based estimates provided in Kossin et al. (2013), who find that lifetime maximum TC intensities in the SIO have increased by about 4.6 m/s over the period 1982-2009 (1.7 m/s per decade), which corresponds to 8.5% of TC Idai's maximum intensity. If this rate of increase is linearly extrapolated to 2019, it results in an increase of about 6.3 m/s (11.6%). Since the rate of increase has likely risen along with surface warming, and since our period of reference extends back to 1973 rather than 1982, a value of 12% might be a safer assumption for comparing the results of Kossin et al. (2013) with our own estimate. To quantify the effect of uncertainty in the estimate of TC intensity change, we conduct two sensitivity experiments, with counterfactual intensity lower than factual by 8.5% and 12%, respectively, reflecting the SOI estimate of Kossin et al (2013) both directly and when extrapolated for comparability with our own estimate.

We note that lower rates of change have been found in climate model-based studies. Knutson et al. (2020) find a 6% increase in maximum intensity of SIO TCs per 2°C global mean surface warming. When applied to the historical increase in global mean surface temperatures of 1.1°C, this would yield an increase of 3.3%. While these climate model estimates are important both for assessing future changes and for understanding the underlying mechanisms of observed trends, the remote-sensing based trend estimates are more relevant for informing the construction of the counterfactual in our study."

Results (lines 760-767):

"Besides our central TC intensification assumption of 10%, we also examine two alternative assumptions of 8.5% and 12% intensification, respectively, for the most plausible tide choices ("max"). The spread among the intensification scenarios is rather small, with median relative changes varying between 2.9% and 3.7%. This translates to median estimates of 35,300 and 44,600 affected people, or 13,400 and 16,900 displacements, respectively (Table 1 and Table S2). In contrast, the difference between the highest (4.0%) and lowest values (2.2%) is larger. In absolute terms, this means a range of between approximately 27,400 and 48,200 affected people, or 10,400 and 18,200 displacements."

- ○ Lines 122-124: Why these increases in sea levels?

We have provided an explanation about our choice of sea level increases in section 3.1, which previously appeared later in the manuscript. The increases are based on the work of Strauss et al. (2021), which we justify in the manuscript to be applicable for the coast of Mozambique (lines 210-216):

"... Given that these regional estimates are close to the global mean estimate by the IPCC, we assume that total SLR near Beira is the same as the global mean, a comparable approach as by Irish and colleagues (Irish et al., 2014). In order to exclude trends induced by natural variability, particularly in sea level contributions from glaciers and ice sheets, we use estimates of global mean sea level rise attributable to anthropogenic climate change for 1900–2012 from Strauss and colleagues (Strauss et al., 2021). Their ensemble estimate is 6.6 to 17.1 cm, which we use to define counterfactual sea level parameters for the coastal flood model."

- ○ Line 131: Why 9 arc sec resolution? What are the limitations when merging these datasets (which are known to have large offsets)?

Thank you for pointing out this caveat. We have added the following:

"... at a spatial resolution of 9 arcseconds (approximately 300 m). This resolution is the highest resolution where we were able to obtain numerically stable results from GeoClaw. We note that no harmonization has been applied to make up for disagreements between the different DEM products so that the transition from CoastalDEM topography to SRTM 15+ bathymetry can be steep."

- ○ Lines 137-139: How is it possible that agreement is highest under the maximum tide assumption, followed by the no tidal adjustment assumption? I would assume that agreement would be second best with the mean tidal assumption.

We agree that this appears to be counterintuitive based on our initial description of the tide assumptions. The run with no tidal adjustment assumes sea levels according to the monthly mean extracted from the gridded satellite altimetry product. In contrast to that, all astronomical tidal adjustments use sub-monthly astronomical tidal information according to the time of landfall. A priori, there is no simple ordinal relationship between the run without tidal adjustment

and the runs with astronomical tidal adjustments. We modified the wording in the main text to avoid misunderstandings: "...and found the best agreement of maximum surge heights for the GeoClaw run with the maximum astronomical tide assumption, closely followed by the run assuming the monthly mean sea level (no tidal adjustment) (Supplementary Figure S1)."

○ Lines 154-156: FloodScan: What is the resolution? What does "small flood sensitivity" mean?

The FloodScan resolution is 90 m, as explained in lines 348-349:

"MERIT DEM provides a roughly 90 m resolution global layer derived from multiple space-based sensors to minimize elevation errors. ... Observed flood extents for TC Idai were obtained from Atmospheric and Environmental Research & African Risk Capacity's accumulated 2-tier standard flood extent depiction FloodScan product from 2019-03-01 to 2019-03-31 using the MERIT DEM resolution."

We revised the sentence (and split in two) to clarify the small flood sensitivity (lines 368-372):

"Originally developed for applications in Africa, this FloodScan algorithm relies on satellite based low-resolution passive microwave data and was designed to capture national-scale events. To accomplish this, the algorithm minimizes false-positives, making the algorithm more prone to false-negatives and less sensitive to events with smaller spatial extent and urban floods (Galantowicz and Picton, 2021)."

○ Lines 159-162: The description of the RICorDE algorithm is unclear to me.

We replaced the description of RICorDE with a more detailed one:

"RICorDE is a tool developed in pyQGIS for post-event analysis of fluvial flood events using inundation masks derived from space-based observations. RICorDE first generates a Height Above Nearest Drainage (HAND) grid followed by an inundation correction phase and a water surface level (WSL) calculation phase. As part of pre-processing, the HAND grid is obtained using WhiteboxTools' ElevationAboveStream (Lindsay, 2014) from the permanent surface water layer and the DEM. In the first phase of RICorDE, the observed flood extents are hydraulically corrected to account for under-predictions using the permanent surface water layer and over-predictions using a HAND-derived inundation representing the upper quartile of possible flooding extents.  In the second phase, HAND values sampled from the inundation shoreline are used to produce an interpolated WSL grid using WhiteboxTools' CostAllocation algorithm (Lindsay, 2014). Finally, gridded water depths are obtained from this WSL grid through subtraction with the DEM. RICorDE is explained in detail in the tool publication (Bryant et al., 2022) and the source code can be accessed online (https://github.com/cefect/RICorDE_pub)."

- Section 2.1: Would it be possible to validate the coastal flood modeling with observed floodplains derived from satellite imagery, similar to the work of Tellman et al. 2021 (who unfortunately do not cover TC Idai)?

Thank you for raising this point, which is crucial regarding the credibility of the whole flood risk analysis. We got access to satellite imagery by AER which is used in this study (description on the webpage[1]). However, it is no suitable for validation of the coastal flood model as we now elaborate more in detail in the manuscript (lines 902-925):

"Similarly, no suitable validation data for the coastal flood simulations is available. According to the AER description, the used products "depict large scale, inland river flooding well but are less likely to depict flooding in smaller floodplains and near coastlines". We have hence opted to not choose the AER product as the sole coastal flood hazard estimate nor as validation dataset for the flood extent from our coastal flood model. A flood risk screening for Beira (van Berchum et al., 2020) showed that simulated flood extent for a 10-year rainfall event plus a 10-year coastal surge event covers most parts of the Central and Munhava city districts of Beira (South-Eastern city districts). In contrast, the satellite imagery by AER shows only little flooding in this area, while it is assumed that flooding by TC Idai exceeded an average recurrence interval of 10 years. For example, Emerton et al. (2020) show that GloFAS flood forecasts indicated a 100% probability of exceeding the severe flood alert threshold (20-year return period) for TC Idai at the Pungwe River (Emerton et al., 2020). Furthermore, newspaper photographs (Bergensia, 2019) show flooding in the Area de Baixa part of Beira (Western district of Beira), which was only partially flooded according to the satellite imagery. The AER product thus likely underestimates flood extent, which  may be explained by cloud obscurement or failure in automatic flood detection due to, for example, flooding in densely populated areas, or the satellite passing over some time after the peak flooding when water levels have already receded."

We validated water levels at a virtual tide gauge station off the coast of Beira with simulated water levels from the Global Tide and Surge Model (GTSM) (Dullaart et al., 2021; Muis et al., 2020) to determine the optimal tide assumption, as laid out in the manuscript. Apart from this analysis, no other dataset is available to our knowledge which would allow for a validation of GeoClaw for this event. We discuss the limitations and uncertainties regarding this matter now also more in depth in the Discussion section of the manuscript (lines 927-935):

"Furthermore, the coastal flood modeling framework does not incorporate any astronomical tidal dynamics. Because there are no tide gauge records available in the region, we were only able to compare the model's surge heights to  the state-of-the-art Global Tide and Surge Model (GTSM). For the derived flood maps, there were no observational benchmarks available for validation. Moreover, the model is not able to take the interaction of the coastal surge with increased river discharge at the estuaries into account. In some cases, this interaction has been shown to influence water levels in a nonlinear way (Bilskie and Hagen, 2018). Another source of uncertainty is again the DEM, in particular the transition from topographic to bathymetric data at the coast lines."
* * *
[1] https://www.aer.com/weather-risk-management/floodscan-near-real-time-and-historical-flood-mapping/

- I am wondering why certain flood depth thresholds are used to calculate the displacement vulnerability ratios instead of determining this threshold from the data, by summing up the population per flood depth increment (starting from the highest depth) until the affected population equals the observed displaced population. In my understanding the minimum flood depth at which people start being displaced could then be derived. Subsequently, this flood depth could be applied to the counterfactual scenarios as well.

Thank you for bringing up this idea. We used a flood depth threshold of 100 cm, used as a simplifying assumption in previous literature (Kam et al. (2021); Custer and Nishijima (2015)). Additionally, we tested the sensitivity to this choice by using alternative thresholds of 10 cm and 50 cm, drawing on Smith and Cox (2010) and the UK Environment Agency (2019). The assumption when using such thresholds is that at a given flood depth, the risk of severe damages to, or even destruction of, residential buildings and other infrastructure typically becomes so large that people *may* be forced to flee. At the same time, it is assumed that the number of people that *actually* become displaced then depends on additional factors, both physical (e.g. the ability of the particular buildings affected to withstand even large flood depths) as well as political and socio-economic (e.g. the capacity of local authorities to issue effective warnings and put in place flood-defense measures, or the capacity of affected people to protect their homes). These factors are not generally known, and may vary between locations and populations. Their aggregate effect is reflected in the ratio between people displaced and people affected - the specific vulnerability observed for a given event. Thus, the link between flood hazard and displacement is "soft" in the sense that it is mediated by the local vulnerability.

An alternative assumption, which you allude to, would be that there is an (event-specific) flood-depth threshold below which there is no displacement, and above which people become displaced regardless; that is, a "hard" link between flood hazard and displacement. In this case, the flood-depth threshold can be calculated directly in the way you describe. When we do this for TC Idai, we arrive at a threshold of about 400 cm (similar for all tide assumptions), for which the modeled number of affected people approximately equals the number of observed displacements. This value is very high in comparison with the thresholds cited above, and we believe it is implausible that displacement occurred only in locations inundated by 4 meters or more.

Nevertheless, this is an insightful analysis, and we have added the following discussion in the Methods section of the manuscript (lines 488-505):

"We point out that the use of predefined flood thresholds implies the assumption that at a given flood depth, the risk of severe damages to, or even destruction of, residential buildings and other infrastructure typically becomes so large that people *may* be forced to flee. The number of people that *actually* become displaced then depends on additional physical, political and socio-economic factors, which may vary between local contexts and are not generally known. Their aggregate effect is reflected in the specific vulnerability factor $v_{i,h}$. In other words, the link

between flood hazard and displacement is "soft" in the sense that it is mediated by the local vulnerability. An alternative assumption would be that there is an (event-specific) flood-depth threshold below which there is no displacement, and above which people become displaced regardless; that is, a "hard" link between flood hazard and displacement. In this case, the flood-depth threshold could be derived directly from the data, as the depth level at which the calculated number of affected people equals the reported number of displacements. When we do this for TC Idai, we obtain a threshold of about 400 cm (similar for all tide assumptions; Supplementary Table S1), for which the modeled number of affected people approximately equals the number of observed displacements. This value is very high in comparison with the thresholds cited further above, and we believe it is implausible for displacement to occur only in locations inundated by 4 meters or more. This exercise therefore lends further justification for the "soft link" approach."

- I am generally concerned about conducting a local-scale study with global-scale datasets. Would it be possible to enrich the data and assumptions with local data (e.g. on population, elevation) and literature (e.g. of reported land subsidence, flood extents and depths) as well as with analysis of satellite imagery (e.g. to derive flood extents for validation)? While I am well aware of the data limitations in the study region, location-specific information may be available or, if lacking, should be discussed in more detail. If the use of global data is intentional (e.g. to ensure reproducibility in other regions), this should be discussed as well.

We fully understand your concern about using global-scale products for local-scale studies. However, the use of global data and globally applicable models has advantages, as you already allude to the virtual global employment of our approach.

Nevertheless, we conducted detailed research on the availability of local-scale hazard information (water level gauges, geolocated imagery, etc.), environmental data (DEM, land subsidence) or socio-economic data (displacement, population distribution), however, almost no local information is available. For example, as pointed out in the manuscript, the nearest tide gauge is located 448 km south of Beira. We address this challenge by controlling for the sensitivity to the choice of SLR, tide assumption, TC intensification (as part of the revision), and critical water level thresholds. We have added the following parts to the introduction and discussion:

Introduction: "Moreover, Mozambique, like many other countries, is characterized by limited availability of in-situ observational data and a lack of calibrated, local-scale inundation models. We use remote-sensing data and a globally applicable modeling framework to characterize flood exposure during TC Idai; reported displacement data is retrieved from the Global Internal Displacement Database (GIDD). Our approach is thus transferable to other cases in virtually all relevant countries."

Discussion: "....Our approach allows for a virtual global employment to assess the impacts of climate change on other under-researched regions, guiding future adaptation, humanitarian and disaster relief efforts."

- The entire section has a shift in tenses. Especially from section 2.2 to 2.3, past tense shifts to present tense in most instances.

We apologize for the shift in tenses and have corrected the corresponding verbs.

Results

- As pointed out above, section 3.1 does not fit in the results section as it provides further detail on the modeling assumptions. I therefore suggest moving this section to the methods.

Done, thank you.

- Line 298: what does p = 0.06 stand for?

We refer to the p-value, testing for a statistically significant relationship between the predictor variable and the response variable. p=0.06 means a 6% probability of the measured relationship occuring by chance. We deleted this detail to avoid confusion, the referenced paper provides further information.

- Section 3.2:
  - The first paragraph largely repeats what has already been stated in the methods. I therefore suggest (re)moving it (line 307-314).

We agree with this comment. We have shortened the first sentence and have removed all other sentences (lines 309-318, original line numbering).

  - I suggest extending lines 321-324, describing the results shown in Figure 3 in more detail, which – in my opinion – should make up the largest part of this section.

Thank you for pointing this out. We have extended our analysis in this section and have also added an additional panel to highlight the differences in simulated flood depth between the factual and counterfactuals runs:

"Beira consists of two major population centers, whereas the southern one is close to the seaside and exhibits a higher population count.

Both factual and counterfactual flood extent covers the southern, highly populated part of Beira (Figure 3c and 3d), the northern population center is only marginally affected. Flood extents are

also similar east of Beira and around the inflow of the Buzi River, located at the opposite of the bay. Only a few isolated locations no longer experience flooding after removing the effects of climate change.

In contrast, differences in simulated flood depth are more pronounced (Figure 3e). Most parts of the southern population center show low flood depth differences, however, partially counterfactual flood depths are up to 80 cm lower. The highest difference in flood depth of up to 140 cm is found between the two population centers of Beira. Despite gridded population data indicating almost no population living in this area, satellite imagery shows the existence of widespread settlements. Flood depth differences outside of Beira are rather low, however, Figure 3c and 3d show that absolute flood depths drop below the critical flood depth of 100 cm over great parts around the west bank of the Pungwe River inflow. Again, also in this region not all settlements are captured by the population dataset.

Overall, it is observable that depth differences are higher in less populated parts, especially in Beira, suggesting that a counterfactual Idai of lower intensity leads to only neglectable changes in displacement. Nonetheless, already small differences in flood depth can cause inundation to drop below the critical flood depths, as shown for the west bank of the Pungwe River. In the next section, we turn to this topic in a numerical way by comparing the number of affected people and displacements between factual and counterfactual simulations."

- ○ Figure 3: It would be useful to have a box in 3a that delineates the zoomed in areas in 3b-d. The color code in c-d is slightly counterintuitive as more intense colors reflect lower flood depths.

Thank you for this suggestion. We have added the zoomed in area to plot 3a.

Regarding the color code: Cyan-like colors are indeed more pronounced than dark blue tones. However, it is a common practice of flood hazard visualization to depict lower flood depth with the former and higher flood depth with the latter. We have hence decided to adhere to this style. Nonetheless, we have changed the background of plot 3b-e to dark gray to increase the visibility.

- ● Section 3.3:
    - ○ The results description is difficult to follow. Might it be better to first describe the factual results, followed by the counterfactual results? Maybe a table would be useful, presenting the results for each SLR assumption, tidal assumption, impact threshold, and hazard (i.e. SLR, wind, both)?

Thank you, this is a good idea. We provide two tables showing the main results (displacement difference (absolute/relative) between factual/counterfactual run) for the SLR, wind, and SLR + wind scenarios. Instead of each SLR assumption, we show the min/median/max results. We furthermore show the results for the two additional intensification assumptions (8.5%, and 12%).

○ It would also be insightful to be walked through the results that are presented in Figure 4. Also, it is unclear to me which change is presented in the bottom panel of Figure 4.

Thank you for this comment. We have updated the results section and also added a table with the main results. We have also added the following line which relates to the bottom panel of Figure 4:

"We compare factual and counterfactual affected people/displacements and compute the absolute relative change based on the counterfactual results, representing the increase in impact due to climate change."

○ Lines 364-367: Is this a hypothetical statement?

We have changed this sentence to remove the hypothetical tone:

"Assuming the same vulnerability factor for displacement due to high wind speed as due to flooding yields 109,200 to 111,500 displacements, or 22.8 to 23.3% of the total displacement, attributable to climate change."

Discussion & conclusion

● While section 4 provides an interesting discussion of technical aspects and limitations of the model assumptions, it lacks more in-depth discussion of the uncertainties stemming from the modeling approach and assumptions as well as the input data used, which are important to contextualize the results. More systematic discussion of these uncertainties would provide interesting insights.

Thank you very much for pointing this out. We agree and have added the following four paragraphs to the discussion to address uncertainties regarding the underlying inland and coastal flood estimations, population count and displacement data (lines 891-950):

[revised manuscript text omitted]

- Furthermore, the discussion largely lacks reflection on the implications of the work in terms of research and policy-making. In the abstract, the authors state that the study "provides a blueprint for event-based displacement attribution" (line 27) which is not elaborated further. Also, the authors state that the storyline approach "raises risk

awareness in a more tangible way" (line 455), which I fully agree with; however, reflections on how to use the results of this work to raise awareness (both in society and in policy-making) are missing. In my opinion, such reflections would add substantial value to the messages conveyed with this research.

Thank you for highlighting this. We have added the following passage regarding policy implications (lines 1004-1036):

"The responsibility for managing and reducing displacement risk lies primarily at the national and provincial level, but often local authorities, organizations, and communities respond to displacement disasters (Hollinger and Sienkevych, 2019). Demonstrating quantitatively how climate change affects the societal risks associated with natural hazards may play an important role in raising awareness, with different types of stakeholders, to the changing nature of such risks. It may also incentivize governments to step up their efforts both in terms of planning and investing into adaptation measures, and rapidly mitigating greenhouse gas emissions. The storyline approach is particularly suited for highlighting the risk-amplifying effects of climate change in a tangible and accessible way, based on a well-known event in the recent past (van den Hurk et al., 2023). Estimates of the costs of displacement additionally highlight the adverse economic aspects of climate change (Desai et al., 2021); average costs have been put at $310 per displaced person per year, though actual costs are heavily dependent on the country and duration (days/weeks to years) (IDMC, 2019). Only 50.7% of the required Mozambique Humanitarian Response Plan 2019 of US$m 620.5 was funded, demonstrating that climate change poses an additional burden to insufficiently equipped financial aid resources."

- Reflections on needs and/or plans for future work would be useful (which also connects to my previous comment). This could include points such as (not exhaustive): Can/Will this approach be used in other contexts? Which aspects of the work can be improved in future work? How may future climate change affect displacement risk, considering both changes in climate as well as changes in socioeconomic conditions.

We agree that such reflections are necessary. We have added the following section regarding the future work:

"Our impact attribution is built on global-scale datasets and models, which could be employed in other relevant locations. Despite the discussed limitations and uncertainties inherent to this approach, displacements could be similarly attributed to climate change for other major TCs that occurred in data- and model-scarce regions, such as Typhoon Haiyan (Philippines; 4.1 million displacements) or Cyclone Amphan (India and Bangladesh; combined 4.95 million displacements) (IDMC, 2022). The continuing increase in spatial resolution of global-scale products will eventually allow for more granular displacement risk assessments, which regional authorities could incorporate in urban development plans, zoning regulations or required building codes (IDMC, 2019). … Our approach may hence be extended to large-n impact attribution, using, for example, global counterfactual climate datasets (Mengel et al., 2021)."

Furthermore, one of the most important tasks for future work would involve the attribution of river flooding to climate change, which we have already discussed in the second paragraph of the Discussion section.

We have additionally added a broad outlook on how the counterfactual drivers of our simulations will evolve in the future:

"Anticipating the intensification of tropical cyclones under future global warming (Knutson et al., 2020) calls for enhancing adaptation measures as well as disaster relief and humanitarian aid. The IPCC AR6 projects an additional global increase in mean sea level and surface temperature of 0.28–0.55 m / 1.0°C–1.8°C (SSP1-1.9) and 0.63–1.01 m / 3.3°C–5.7°C (SSP5-8.5) by the end of the 21st century (Chen et al., 2021). Even though these increases may vary between basins, an enhanced displacement risk due to Idai-like TCs needs to be accounted for in the next decades, especially if future changes in exposure due to population growth and urbanization are considered."


Thank you for this comment, the manuscript now includes sections regarding these points, please see below.

More detailed comments are given by line number below.

Abstract

16: 'massive' is a strange adjective to use, particularly in the first sentence.

Removed.

17: 'frequency and intensity of such events is affected by anthropogenic climate change' - statement seems to presuppose the subject of the paper.

Thank you and we apologize for the ambiguous statement. The phrase "such events" was meant to refer to tropical cyclones, not to displacement events. Indeed, there is ample evidence that climate change affects the intensity and frequency of tropical cyclones, in spite of uncertainties around the numbers (as discussed in the Introduction). However, there is little knowledge about the contribution of climate change to displacement induced by tropical cyclones, which is the subject of our study. We have rephrased as follows:

"Extreme weather events, such as tropical cyclones, often trigger population displacement. The frequency and intensity of tropical cyclones is affected by anthropogenic climate change. However, the effect of historical climate change on displacement risk has so far not been quantified."

In response to a related comment by reviewer #1, we have also added a paragraph in the Introduction (lines 74-93) that discusses in more detail the state of knowledge about the climate change effect on tropical cyclones and their societal impacts:

"Historic TCs data records are short and partially inconsistent, making it difficult to determine the degree of intensification over time, despite observed changes in some basins, such as the South Indian Ocean (Knutson et al., 2019; Kossin et al., 2013, 2007; Webster et al., 2005). Moreover, existing TC datasets often focus on maximum wind speed, neglecting coastal and inland flooding which may be the dominant hazards, e.g., as for Hurricane Katrina or Hurricane Harvey (Bloemendaal et al., 2021). Paleo climate records (Lin et al., 2014; Nott and Hayne, 2001) and synthetic TC tracks (Bloemendaal et al., 2022, 2020; Emanuel et al., 2006) can be used to extend TC records. However, sediment availability is limited to a few coastal stretches and the statistical resampling process incorporates only the average observed climatic conditions, respectively, hampering the assessment of global climate change impacts over longer time periods (Bloemendaal et al., 2020). Nonetheless, given that global mean surface air temperature and sea level have already risen above pre-industrial conditions (by about 1.1°C and 0.20 m, respectively (Gulev et al., 2021)), it is likely that recent TC landfalls have caused more severe societal impacts than would be expected without climate change. A probabilistic attribution addressing this topic is limited by the shortness of TC records (Trenberth et al., 2015), and may be additionally affected by multi-decadal variability (e.g., the Atlantic Multidecadal Oscillation) or interannual climate variability (e.g., the El Niño–Southern Oscillation) (Patricola and Wehner, 2018). As a consequence, the portion of TC-induced human displacements attributable to climate change has so far not been quantified."

21-22:   Terms used to describe methodology of 'storm surge modelling' could be detailed more clearly. It is worth defining the actual approach and methodology employed.

We have modified the Abstract to give more detail on our approach (underlined):

"We estimate the population exposed to high water levels following Idai's landfall, using a combination of a 2D hydrodynamical storm surge model and a flood depth estimation algorithm to determine inland flood depths from remote sensing images, for factual (climate change) and counterfactual (no climate change) mean sea level and maximum wind speed conditions."

25:     Are these percentages compared to number of people displaced, or of exposed populations?

We rephrased the sentence:

"Our main estimates indicate that climate change has increased displacement risk from this event by approximately 12,600 - 14,900 additional displaced persons, corresponding to about 2.7 to 3.2%"

Introduction

- Does not refer to the potential impact of climate change on TC storm track. This isn't something that is necessary for inclusion in the study itself, but perhaps should be referred to in context. This is highlighted in the Discussions/Conclusions, but can be stated here as well.

Thank you for pointing this out, we have added a corresponding note to the Introduction:

"Storm track and size are not changed, even though both parameters are subject to the effects of climate change (Knutson et al., 2020, 2019)."

29:     Use of term 'immense' is ambiguous - is there a clearer alternative?

We have replaced this and other vague terms. The sentence referred to here now reads:

"TCs pose a set of societal risks to coastal communities around the world."

32-33:     'massive damages to housing and infrastructure' is again ambiguous. Is there a quantifiable range, or example time period?

Replaced with the following sentence:

"While related monetary losses are high, with an average of US$ 57.2 billion damages every year since 2008 (Guha-Sapir et al., 2022), TCs also displace an average of 9.3 million people every year, with this hazard being responsible for 43% of all weather-related displacements (IDMC, 2022)."

33-34:   Are there additional comparative sources for the estimates of those displaced? This and the Desai et al. (2021) reference are drawing on IDMC reporting, so would benefit from other source corroborating or providing range. Is there an IPCC figure?

We acknowledge your concerns about the displacement estimates. Indeed, an independent source would be desirable to corroborate the IDMC figures, but to our knowledge, there is no alternative dataset that is comparable in quality to the IDMC estimates. IDMC gathers displacement data from various sources including government agencies, UN organizations

(OCHA, UNHCR, etc.), IOM, and others. We further elaborate on the GIDD data in the manuscript, please see the next point.

A definition of what forms of human mobility are considered 'displacement' in the context of this study is required. It is noted that in section '2.4 Displacement' the GIDD source (IDMC, 2022) is described as lacking granularity. The definition of displacement used by IDMC, however, should be outlined.

We fully agree that our initial displacement description lacked a clear definition. We have extended the manuscript with the following displacement definition and also added a description/discussion of the underlying displacement data related to TC Idai:

Methods (lines 424-438)::

"We use displacement data from the publicly accessible GIDD, maintained by the Internal Displacement Monitoring Centre (IDMC, 2022). IDMC follows the definition of displacement provided in the Guiding Principles on Internal Displacement (OCHA, 2004), which states that "[i]nternally displaced persons are persons or groups of persons who have been forced or obliged to flee or to leave their homes or places of habitual residence, ... and who have not crossed an internationally recognized State border". This definition covers permanent displacement, temporary displacement, and pre-emptive evacuations (Gemenne, 2011), all summarized as "displacements" within our study. No granular information is available in GIDD on the type of displacement. Displacement numbers are based on multiple secondary sources, such as IOM, OCHA, or - in the case of TC Idai - the Mozambique National Institute of Disaster Management. The TC Idai event is categorized as a "storm" event, however, no information is given on how many of the displacements were caused respectively by flooding, strong winds, or a combination of both. Because of the extensive flooding observed in the wake of Idai's landfall, and humanitarian reports often focused on flooding (ReliefWeb, 2019a), we assume in our main analysis that all displacements are caused by flooding (either coastal or inland)."

Discussion (lines 940-950):

"The total number of displacements is furthermore not specifically categorized by hazard type, which reflects the multivariate (wind, rain and flood) compound characteristic of TCs hazards (Zscheischler et al., 2020). However, this impedes the attribution of coastal flood-induced displacements. Furthermore, the GIDD estimates include different forms of displacement, such as forced displacement or pre-emptive evacuations, with the latter potentially accounting for a substantial proportion (McAdam, 2022). This poses far-reaching implications for displacement risk modeling, as evacuations may already be triggered by lower flood depths, or by early warnings of an impending hazard, which may not materialize in the expected manner, or may not cause the level of destruction that would lead to a corresponding magnitude of forced displacement."

42-43: Emmanuel, 1987 seems like quite an 'old' reference. Following papers relevant for increasing TC intensity potentially:

- Emanuel, K. (2005). Increasing destructiveness of tropical cyclones over the past 30 years. Nature, 436(7051), 686–688. https://doi.org/10.1038/nature03906
- Emanuel, K. A. (2013). Downscaling CMIP5 climate models shows increased tropical cyclone activity over the 21st century. Proceedings of the National Academy of Sciences, 110(30), 12219– https://doi.org/10.1073/pnas.1301293110

Thank you for the suggested references, we have added them.

46-48: 'substantially' unnecessary if IPCC figures are included in parenthesis - better to just cite the IPCC reference

Done.

**Methods**

**Coastal Flood Modelling**

108-110: Why does the study focus on an equivalent 10% increase in intensity due to changing climate?

*"For the counterfactual scenarios with modified TC intensity, we 107 multiplied all wind speed values along the track by a scalar factor of 0.9 (for a decrease of 108 10% in intensity). The central pressure at each track position is increased by 0.1 times the 109 difference between central pressure and environmental pressure."* (Mester et al., 2023)

The Knutson et al. (2020) paper cited suggests that the "*…Fifteen individual scaled global estimates are all positive, with a mean and median increase (range) of about 5% (1%–10%)*" (Knutson et al., 2020). Therefore the 10% difference that this study ascribes to climate change is at the very upper end of this boundary.

The explanation of this is outlined in 'Results' section, lines 284-301. These sections should perhaps be combined so the explanation / justification of this are presented together. It is noted - as highlighted in the paper - that the 10% change is at the upper estimate of other studies. Should a number of model runs at different degrees of change be presented?

Thank you for this comment and the suggestions! We have moved the first part of the Results section into the Methods section, such that the construction of the counterfactuals is explained first, right before the explanation of the coastal flood modeling. We agree that this structure makes it easier to follow our arguments.

We have also followed your suggestion to include additional model simulations, as explained below.

The value of 10% is based on our own estimate of the trend in Southern Indian Ocean (SIO) tropical cyclone maximum wind speeds, using IBTracs data. It is, however, very much in line with the remote sensing-based estimates provided in Kossin et al. (2013), who find that lifetime

maximum TC intensities in the SIO have increased by about 4.6 m/s over the period 1982-2009 (1.7 m/s per decade), which corresponds to 8.5% of TC Idai's maximum intensity. If this rate of increase is linearly extrapolated to 2019, it results in an increase of about 6.3 m/s (11.6%). Since the rate of increase has likely risen along with surface warming, and since our period of reference extends back to 1973 rather than 1982, a value of 12% might be a safer assumption for comparing the results of Kossin et al. (2013) with our own estimate. Thus, our choice of 10% is actually lower than the extrapolation of these published results, which are also extensively cited in the overview paper by Knutson et al. (2019).

In contrast, the Knutson et al. (2020) study is based on climate models rather than remote observations. The value of 5% per 2°C global mean surface warming refers to the median global estimate in that study; for the SIO, we extract a value of about 6% per 2°C from their results. When applied to the historic increase in global mean surface temperatures of 1.1°C, this would yield an increase of 3.3%. While these climate model estimates are important both for assessing future changes and for understanding the underlying mechanisms of observed trends, we believe the remote-sensing based trend estimates are more relevant for informing the construction of the counterfactual in our study. While the analysis in Kossin et al. (2013) is based on only about 30 years of data, our own analysis of almost 50 years worth of data (1973-2019) leads to a very similar result.

To explore the effects of the undeniable uncertainties in these estimates, we have added to our central run with 10% TC intensity increase, two runs assuming 8.5% and 12% increase respectively, reflecting the SOI estimate of Kossin et al (2013) both directly and when extrapolated for comparability with our own estimate:

Methods (lines 235-259):

"This value is in line with the remote sensing-based estimates provided in Kossin et al. (2013), who find that lifetime maximum TC intensities in the SIO have increased by about 4.6 m/s over the period 1982-2009 (1.7 m/s per decade), which corresponds to 8.5% of TC Idai's maximum intensity. If this rate of increase is linearly extrapolated to 2019, it results in an increase of about 6.3 m/s (11.6%). Since the rate of increase has likely risen along with surface warming, and since our period of reference extends back to 1973 rather than 1982, a value of 12% might be a safer assumption for comparing the results of Kossin et al. (2013) with our own estimate. To quantify the effect of uncertainty in the estimate of TC intensity change, we conduct two sensitivity experiments, with counterfactual intensity lower than factual by 8.5% and 12%, respectively, reflecting the SOI estimate of Kossin et al (2013) both directly and when extrapolated for comparability with our own estimate.

We note that lower rates of change have been found in climate model-based studies. Knutson et al. (2020) find a 6% increase in maximum intensity of SIO TCs per 2°C global mean surface warming. When applied to the historical increase in global mean surface temperatures of 1.1°C, this would yield an increase of 3.3%. While these climate model estimates are important both for assessing future changes and for understanding the underlying mechanisms of observed

trends, the remote-sensing based trend estimates are more relevant for informing the construction of the counterfactual in our study."

Results (lines 760-767):

"Besides our central TC intensification assumption of 10%, we also examine two alternative assumptions of 8.5% and 12% intensification, respectively, for the most plausible tide choices ("max"). The spread among the intensification scenarios is rather small, with median relative changes varying between 2.9% and 3.7%. This translates to median estimates of 35,300 and 44,600 affected people, or 13,400 and 16,900 displacements, respectively (Table 1 and Table S2). In contrast, the difference between the highest (4.0%) and lowest values (2.2%) is larger. In absolute terms, this means a range of between approximately 27,400 and 48,200 affected people, or 10,400 and 18,200 displacements."

**Displacement**

195 onwards: Necessarily, the relationship between the impact of the TC and displacement is artificially deterministic. Within the context of this model, however, this approach seems reasonable.

Thank you for this comment, we now address this point more in detail (lines 442-451):

"Our modeling approach assumes an artificially deterministic link between the TC hazard and displacement, which is adequate in the context of the factual-counterfactual approach where only one parameter - storm surge hazard - is modified while everything else, including vulnerability, is held constant. In general, the relationship between climatic events, pre-existing socio-economic conditions, and displacement is complex and only partially understood (Cattaneo et al., 2019; UK Government Office for Science, 2011). In other words, our study addresses the question of how many displacements might have occurred in a different climate but with the same vulnerability as observed; it does not address the question of how this vulnerability came about."

**Results**

**Counterfactuals**

242 onwards: This section seems to be more focused on methodology rather than results. Can it be relocated within the paper?

Thank you for this suggestion. We shifted this section to the Methods.

323: *"Notably, in a world without climate change, the area inundated by 100 cm or more 323 is dramatically reduced."* Although this is demonstrated in comparison of fig.3 (c) and (d), it seems that the areas where there is most significant difference in inundation (e.g. western bank of main flood valley) are those that are least densely populated according to fig.3 (b). This perhaps should be highlighted?

Thank you for this remark. We have extended the description of Figure 3 and have also added an additional panel to highlight the differences in simulated flood depth between the factual and counterfactuals runs:

"Beira consists of two major population centers, of which the southern one is close to the seaside and exhibits a higher population count.

Both factual and counterfactual flood extent covers the southern, highly populated part of Beira (Figure 3c and 3d). The northern parts of the city are only marginally affected. Flood extents are also similar between factual and counterfactual simulations in the areas east of Beira and around the inflow of the Buzi River, located on the opposite side of the bay. Only a few isolated locations no longer experience flooding after  removing the effects of climate change.

In contrast, differences in simulated flood depth are more pronounced (Figure 3e). Counterfactual flood depths are up to 80 cm lower than factual flood depth in some parts of the southern city center. The highest difference in flood depth, of up to 140 cm, is found between the northern and southern population centers of Beira. Flood depth differences outside of Beira are rather low, however, Figure 3c and 3d show that absolute flood depths drop below the critical flood depth of 100 cm over great parts around the west bank of the Pungwe River inflow."

Can the area described in figs.3 (b-d) be outlined on fig.3 (a) alongside that of wider 'area of interest' so it is easier to see the relationship between (a) and (b-d)?

We agree that this provides more insight on the location of Figure (b-d, now b-e). We have updated Figure 3 (a) and inserted the according area of interest.

**Displacement**

337-344: The finding that it is a combination of TC intensity and tide assumptions that have a more significant impact on displacement in comparison to SLR is notable and could be highlighted in the Abstract

Done:

".... The effect of wind speed intensification is larger than that of sea level rise …."

340 onwards: The results described are not always clear and therefore can perhaps be better expressed.

The baseline for the study is the 'factual' displacement of 478,000 people. The counter factual scenarios derived from the model suggest that fewer people would be displaced without contribution of climate change.

Although the line "*The coupled effect of higher wind speeds and higher sea level increases the number of affected people and displacements by up to 43,300 and 16,500 (maximum tide) 350 and 44,300 and 17,100 (monthly mean), respectively.*" is accurate in terms of relative difference

to other counterfactuals, it is potentially unclear as it refers to an '*increases in the number of affected people and displacements*'.

For consistency, it may be clearer to use 'relative change' as the base metric throughout, as described in the 'bottom' panel of fig.4. This would clarify the relative differences between the scenarios in relation to the 'factual' baseline.

Thank you for this comment. We have revised the description of our results to make sure that numbers always refer to the difference between a given counterfactual and the factual baseline, and not (potentially ambiguous) to differences between two different counterfactuals. We have also added the following sentence to explain the "relative change" used in the Results:

"We compare factual and counterfactual affected people/displacements and compute the absolute relative change based on the counterfactual results, representing the increase in impact due to climate change."

**Conclusion**

387: The conclusion focuses on 'about 17,000' additional displacements. This should be given as a range reflecting the different tidal assumptions.

Thank you for this comment. As the validation analysis shows the best agreement for the maximum and monthly mean tides, we concentrate in the discussion on these two assumptions. We have altered the sentence to:

"Our modeling study of TC Idai suggests that climate change may have induced between 12,600 (2.7%; lowest estimate under the no tide assumption) and 14,900 (3.7%; highest estimate under maximum tide assumption) additional displacements from this one event."

Note that these numbers are somewhat lower than the original "about 17,000" due to the correction of a software issue, as explained at the top of this letter.

392 onwards: Points regarding the potential impacts of inland flooding and high wind speeds on displacement are noted and provide useful context for the limitations of the study.

What is lacking, however, is an acknowledgement of the complexity of the relationship between environmental impacts, pre-existing socio-economic conditions and displacement. This is particularly important as what forms of human mobility constitute 'displacement' are not defined within the paper. Although the assumptions on which the relationship between the impact of TC and displacement are valid in the context of this study, the assumptions themselves should be acknowledged. For reference:

- Cattaneo, C., Beine, M., Fröhlich, C. J., Kniveton, D., Martinez-Zarzoso, I., Mastrorillo, M., Millock, K., Piguet, E., & Schraven, B. (2019). Human Migration in the Era of Climate Change. Review of Environmental Economics and Policy, 13(2), 189–206. https://doi.org/10.1093/reep/rez008

- Foresight: Migration and Global Environmental Change (Final Project Report, p. 234). (2011). The Government Office for Science. https://assets.publishing.service.gov.uk/government/uploads/system/uploads/attachment _data/file/287717/11-1116-migration-and-global-environmental-change.pdf

Thank you for pointing this out and providing these references, which we have incorporated into the manuscript (see responses above). We have added the displacement definition according to IDCM, and a description of the GIDD data. We also address now the complexity between displacement, climatic shocks and pre-conditioning factors (lines 424-438):

"We use displacement data from the publicly accessible GIDD, maintained by the Internal Displacement Monitoring Centre (IDMC, 2022). IDMC follows the definition of displacement provided in the Guiding Principles on Internal Displacement (OCHA, 2004), which states that "[i]nternally displaced persons are persons or groups of persons who have been forced or obliged to flee or to leave their homes or places of habitual residence, ... and who have not crossed an internationally recognized State border". This definition covers permanent displacement, temporary displacement, and pre-emptive evacuations (Gemenne, 2011), all summarized as "displacements" within our study. No granular information is available in GIDD on the type of displacement. Displacement numbers are based on multiple secondary sources, such as IOM, OCHA, or - in the case of TC Idai - the Mozambique National Institute of Disaster Management. The TC Idai event is categorized as a "storm" event, however, no information is given on how many of the displacements were caused respectively by flooding, strong winds, or a combination of both. Because of the extensive flooding observed in the wake of Idai's landfall, and humanitarian reports often focused on flooding (ReliefWeb, 2019a), we assume in our main analysis that all displacements are caused by flooding (either coastal or inland)."

And in lines 944-950:

"Furthermore, the GIDD estimates include different forms of displacement, such as forced displacement or pre-emptive evacuations, with the latter potentially accounting for a substantial proportion (McAdam, 2022). This poses far-reaching implications for displacement risk modeling, as evacuations may already be triggered by lower flood depths, or by early warnings of an impending hazard, which may not materialize in the expected manner, or may not cause the level of destruction that would lead to a corresponding magnitude of forced displacement."

There are also significant uncertainties and limitation associated with the methodology of deriving and estimating 'factual' displacement figures by IDMC and these should also be acknowledged and discussed.

Thank you. We have addressed this point also with the previous answers, which can be found in the manuscript in lines 424-438 and 944-950..

Although in a later section (433-441) socio-economic conditions are referred to in terms of a potential change in vulnerability, this is separate point relating to future changes with respect to exposure indices.

Thank you; indeed, the role of socio-economic context even for determining present-day displacement risk was not sufficiently acknowledged. As mentioned in the response to your comment above, we have now added some discussion of this in the Methods section (lines 442-451):

"Our modeling approach assumes an artificially deterministic link between the TC hazard and displacement, which is adequate in the context of the factual-counterfactual approach where only one parameter - storm surge hazard - is modified while everything else, including vulnerability, is held constant. In general, the relationship between climatic events, pre-existing socio-economic conditions, and displacement is complex and only partially understood (Cattaneo et al., 2019; UK Government Office for Science, 2011). In other words, our study addresses the question of how many displacements might have occurred in a different climate but with the same vulnerability as observed; it does not address the question of how this vulnerability came about."

And in the Discussion section (lines 985-987):

"Assessments of future risks - or of past impacts - should not only take into account the intensification of physical hazards, but also changes in exposure (Kam et al., 2021); as well as potential changes in vulnerability due to social, economic, or technological developments. For instance, TC-related displacements depend not only on the damage to housing, but also on other factors such as government responsiveness or poverty levels (Cissé et al., 2022)."

---

## Referee Report (RR1)

The changes made to the manuscript "Human displacements from tropical cyclone Idai attributable to climate change" have improved the paper structure and coherence substantially. There are still several open issues that need to be addressed to make the manuscript acceptable for publication:

1. Thank you for adding a sentence to the abstract. Might it be possible to quantify the effect of wind intensification compared to sea-level rise?

2. Figure 3: In the extended text describing the figure on section is missing that is referenced in the response to one of my comments:

   "Overall, it is observable that depth differences are higher in less populated parts, especially in Beira, suggesting that a counterfactual Idai of lower intensity leads to only neglectable changes in displacement. Nonetheless, already small differences in flood depth can cause inundation to drop below the critical flood depths, as shown for the west bank of the Pungwe River. In the next section, we turn to this topic in a numerical way by comparing the number of affected people and displacements between factual and counterfactual simulations."

   Should this be added to the lines following line 436? Also, I was wondering whether the differences in flood depths in less populated parts might be related to the DEM data as these surface models are known to overestimate elevations in densely populated locations due to the fact that the first reflected signal is registered by the satellite, thereby resulting in lower flood depths in dense urban settings?

   Furthermore, as Fig. 3e shows the difference between 3c and 3d, might it be useful to use a different color scale for depicting these differences?

3. "AER" (l. 583) is mentioned, but not defined in the manuscript. Are those the remote sensing data?

4. L. 611: the sensitivity of results to the choice of population data should be elaborated here, particularly considering the two studies that are referenced along with this statement: which aspects may affect exposure results?

5. L 699-700: the authors allude to changes in population and urbanization in driving future risk. This statement would profit from more specific context, supported by population and urbanization projections for the country under different scenarios (e.g. SSPs), particularly as the authors refer to these scenarios in the preceding lines.

6. My biggest concern are still the uncertainties inherent in this study, stemming from the data as well as the modeling approach and assumptions. Although the authors have done a great job in discussing these uncertainties, I am unsure about the confidence in these findings. Is it really possible to attribute these migrants to climate change? The authors state rather specific numbers which may give a false sense of accuracy, e.g. 2.7-3.2 %. Furthermore, these uncertainties should be stated clearly and discussed explicitly in the abstract as well as the Discussion and conclusion section.

---

## Referee Report (RR2)

**2ND REVIEW NOTES - MESTER ET AL., 2023 - "HUMAN DISPLACEMENTS FROM TROPICAL CYCLONE IDAI ATTRIBUTABLE TO CLIMATE CHANGE"**

Mester, B., Vogt, T., Bryant, S., Otto, C., Frieler, K., & Schewe, J. (2023). Human displacements from tropical cyclone Idai attributable to climate change [Preprint]. Sea, Ocean and Coastal Hazards. https://doi.org/10.5194/egusphere-2022-1308

**Overview**

This study aims to model and identify the 'excess' population displacement triggered by tropical cyclone Idai that can be attributed to climate change. By utilising a 'storyline approach' the study compares actual recorded displacement against estimated levels of displacement derived from counterfactual scenarios of mean sea level and maximum wind conditions without contributions of climate change. The conclusion of the study is that the impact of climate change has increased displacement risk by between 3.1 to 3.5%, corresponding to 16,000 – 17,000 displaced people.

The approach and results are interesting and of value in exploring the impact of climate change on environmental shocks and stressors, and the resulting affect this will have on patterns of human mobility.

The revisions that have taken place since the first review have significantly strengthened the paper. The assumptions on which the modelling approach is based are clearly outlined. The results of the study are much more clearly elucidated, alongside the limitations of results. This contextualises the author's use of a 'storyline approach' as opposed to, for example, a probabilistic approach and highlights the utility of this framing in terms of raising awareness of potential risks.

This is now a good paper and worthy of publication. This recommendation is made both in terms of the results themselves and the blueprint that it offers for future studies. As the authors highlight, the use of counterfactuals and 'storyline' approaches have potential in event-based displacement attribution.

A few minor textual changes and edits are suggested below, including an update of a hyperlink.

**Astract**

26            *"corresponding to about 2.7 to 3.2%."* Can these percentages be clarified? I presume they are of recorded / estimated actual displacees.

**1  Introduction**

32            missing space 'TCs pose'

45            Suggest breaking sentence to improve clarity: *'... very intense TCs (category 4-5 on the Saffir-Simpson scale). This is...'*

**2  Methods**

**2.1 Counterfactuals**

143            Mean sea level rise is expressed here in centimeters ('23cm') whereas later in the paragraph the IPCC rates are expressed in mm yr$^{-1}$. Using millimeters in this instance (230mm, for example) is suggested.

**2.3 Inland Flood Depth Estimation**

304          The location of the source code referenced has changed and the link should be updated to https://github.com/NRCan/RICorDE/tree/main.

**2.5 Displacements**

342-347          This is a useful detail of the limitations of GIDD information

357-362          Again useful section which acknowledges the complexity of driver of displacement which helps place the results of this paper in context.

398-399 & 401  There is an inconsistency between 'flood depth threshold' and 'flood-depth threshold', where the latter is preferred.

**4 Discussions and conclusions**

545-549          This is a clear exposition of the findings.

562          Additional space 'ideally coupled'

674 onwards          Useful framing of the 'storyline approach' in terms of its potential role in raising awareness of risks

697          Hyphen required in 'Idai-like'

---

## Author Response (AR2)

We would like to thank both reviewers for providing further suggestions and comments on the manuscript. Following both reviews, we have revised the manuscript in several ways.

In the following, we respond (blue font) more in detail to every single referee comment (black font). Line references are related to the revised manuscript unless otherwise stated. Underlined text indicates changes within sentences/paragraphs.

Report #1 by Anonymous referee #1

The changes made to the manuscript "Human displacements from tropical cyclone Idai attributable to climate change" have improved the paper structure and coherence substantially. There are still several open issues that need to be addressed to make the manuscript acceptable for publication:

1. Thank you for adding a sentence to the abstract. Might it be possible to quantify the effect of wind intensification compared to sea-level rise?

The isolated effects of wind intensification is twice that of SLR (median assumption). We have changed the sentence accordingly:

Line 27: "The isolated effect of wind speed intensification is double that of sea level rise."

2. Figure 3: In the extended text describing the figure on section is missing that is referenced in the response to one of my comments:

"Overall, it is observable that depth differences are higher in less populated parts, especially in Beira, suggesting that a counterfactual Idai of lower intensity leads to only neglectable changes in displacement. Nonetheless, already small differences in flood depth can cause inundation to drop below the critical flood depths, as shown for the west bank of the Pungwe River. In the next section, we turn to this topic in a numerical way by comparing the number of affected people and displacements between factual and counterfactual simulations."

Should this be added to the lines following line 436?

Thank you for noticing this. We modified this paragraph in the final version of the last revised manuscript, but did not update the response letter accordingly. We apologize for the inconvenience. Following up on your comment below, we have now extended the paragraph to address this topic more in depth.

Also, I was wondering whether the differences in flood depths in less populated parts might be related to the DEM data as these surface models are known to overestimate elevations in densely populated locations due to the fact that the first reflected signal is registered by the satellite, thereby resulting in lower flood depths in dense urban settings?

Thank you, this is a valid point. When we say that the "differences in flood depths" are higher in less populated than in more populated parts, we refer to the differences between factual and counterfactual simulations. Since both simulations use the same DEM, we would assume that any urban-rural biases in the DEM are consistent between the factual and counterfactual simulations. In the most recent revised version of the manuscript, we have improved the formulation to be clear about the kind of comparison that we refer to. Nonetheless, your suggestion about the cause of the flood depth differences is justified and we address this point now as well:

Line 446-452: "In contrast, differences in simulated flood depth are more pronounced (Figure 3e). Counterfactual flood depths are up to 80 cm lower than factual flood depth in some parts of the southern city center. The highest difference in flood depth, of up to 140 cm, is found between the northern and southern population centers of Beira. Flood depth differences outside of Beira are rather low, however, Figure 3c and 3d show that absolute flood depths drop below the critical flood depth of 100 cm over great parts around the west bank of the Pungwe River inflow. Overall, it is observable that depth differences (between factual and counterfactual simulations) are higher in less populated parts, especially in Beira. This could partly result from the fact that digital elevation models tend to overestimate elevation in dense urban settings (Shen et al., 2019), thereby underestimating flood depth and potentially also differences in flood depth between different scenarios, however, this is hard to ascertain given the available data. Nonetheless, local variations in simulated flood depth should be interpreted with care."

Furthermore, as Fig. 3e shows the difference between 3c and 3d, might it be useful to use a different color scale for depicting these differences?

We agree and have changed the color, as shown below.

[Figure]

3. "AER" (l. 583) is mentioned, but not defined in the manuscript. Are those the remote sensing

Data?

Thank you for pointing to this. Atmospheric and Environmental Research (AER) is the provider of the satellite imagery. We have opted to refer to this product in the mentioned section by its name "FloodScan", as done in previous sections, instead of using "AER".

Lines 608-618: "Similarly, no suitable validation data for the coastal flood simulations is available. According to the FloodScan description (Atmospheric and Environmental Research & African Risk Capacity, 2022), the used products "depict large scale, inland river flooding well but are less likely to depict flooding in smaller floodplains and near coastlines". We have hence opted to not choose the FloodScan product as the sole coastal flood hazard estimate nor as validation dataset for the flood extent from our coastal flood model. A flood risk screening for Beira (van Berchum et al., 2020) showed that simulated flood extent for a 10-year rainfall event plus a 10-year coastal surge event covers most parts of the Central and Munhava city districts of Beira (South-Eastern city districts). In contrast, the FloodScan product shows only little flooding in this area, while it is assumed that flooding by TC Idai exceeded an average recurrence interval of 10 years."

4. L. 611: the sensitivity of results to the choice of population data should be elaborated here, particularly considering the two studies that are referenced along with this statement: which aspects may affect exposure results?

Thank you for highlighting this. We now discuss some drawbacks related to the GHS-POP dataset in the manuscript:

Lines 638-648: "Additionally, our analysis may be sensitive to the choice of population dataset (Archila Bustos et al., 2020; Leyk et al., 2019), which may lead to uncertainties regarding our estimated exposure. One of the main error sources for population datasets is related to the areal interpolation methods to disaggregate the population data (Archila Bustos et al., 2020). GHS-POP distributes population only within built-up areas, which has the downside that non-residential areas are simulated as populated as well (Freire et al., 2016). In fact, a comparison with satellite imagery reveals that some areas in Beira are populated which are most likely only commercial or industrial sites. On the other hand, not all settlements are captured by GHS-POP, most likely due to their building type. Nonetheless, GHS-POP is still one of most accurate datasets in estimating and modeling the known population (Archila Bustos et al., 2020), especially in urban contexts (Leyk et a., 2019) as in the case for Beira."

5. L 699-700: the authors allude to changes in population and urbanization in driving future risk. This statement would profit from more specific context, supported by population and urbanization projections for the country under different scenarios (e.g. SSPs), particularly as the authors refer to these scenarios in the preceding lines.

Thank you for this suggestion, we have added the following sentence:

Lines 746-751: "Even though these increases may vary between basins, an enhanced displacement risk due to Idai-like TCs needs to be accounted for in the next decades, especially if future changes in exposure due to population growth and urbanization are considered. Under both SSPs 1 and 5, the population of Mozambique is projected to increase by approximately 8 million, and its urbanization level from about 40% to over 70%, just over the next 30 years (Riahi et al. 2017)."

6. My biggest concern are still the uncertainties inherent in this study, stemming from the data as well as the modeling approach and assumptions. Although the authors have done a great job in discussing these uncertainties, I am unsure about the confidence in these findings. Is it really possible to attribute these migrants to climate change? The authors state rather specific numbers which may give a false sense of accuracy, e.g. 2.7-3.2 %. Furthermore, these uncertainties should be stated clearly and discussed explicitly in the abstract as well as the Discussion and conclusion section.

We thank you for sharing your concerns. We agree that there are several sources of uncertainty related to our study, which must be clearly stated and discussed. In the last revision, we have extended our discussion of uncertainties and modeling assumptions, which now spans eight paragraphs in the Discussion and Conclusions section and, from our perspective, covers all the relevant points; we are glad to hear that you approve of this extended discussion. Nevertheless, we should indeed have highlighted these uncertainties even more. Following your suggestion, we have added a statement about the uncertainties to the Abstract, which also puts into perspective the numbers resulting from our main model estimates and thereby hopefully avoiding a sense of false accuracy; and we have added further statements and explanations in the "Discussion and Conclusion" section; see below.

Regarding the attribution approach: We should have made clear that we do not attribute displacement of individual persons to climate change; this is virtually impossible due to the manifold drivers which shape the vulnerability or even trigger the displacement outcome on an individual level. Our approach amounts to statistically attributing a total number of displaced persons to climate change, similar to e.g. the attribution of TC-related total economic damages (Strauss et al., 2021). In this way, we provide insight into the overall order of impact of global warming on coastal flooding and related displacements. With respect to the challenges of our approach, it is a reasonable advice to put our numbers directly into context of the related uncertainties.

Abstract:

Lines 16-32: "Extreme weather events, such as tropical cyclones, often trigger population displacement. The frequency and intensity of tropical cyclones  is affected by anthropogenic climate change. However, the effect of historical climate change on displacement risk has so far not been quantified. Here, we show how displacement can be partially attributed to climate change, using the example of the 2019 tropical cyclone Idai in Mozambique. We estimate the population exposed to high water levels following Idai's landfall, using a combination of a 2D

hydrodynamical storm surge model and a flood depth estimation algorithm to determine inland flood depths from remote sensing images, for factual (climate change) and counterfactual (no climate change) mean sea level and maximum wind speed conditions. Our main estimates indicate that climate change has increased displacement risk from this event by approximately 12,600 - 14,900 additional displaced persons, corresponding to about 2.7 to 3.2% of the observed displacements. The isolated effect of wind speed intensification is double that of sea level rise. These results are subject to important uncertainties related to both data and modeling assumptions, and we perform multiple sensitivity experiments to assess the range of uncertainty where possible. Besides highlighting the significant effects on humanitarian conditions already imparted by climate change, our study provides a blueprint for event-based displacement attribution."

First paragraph of the Discussion and conclusions:

Lines 561-575: "Our modeling study of TC Idai suggests that climate change may have induced between 12,600 (2.7%; lowest estimate under the no tide assumption) and 14,900 (3.2%; highest estimate under the maximum tide assumption) additional displacements from this one event. This is primarily due to the intensification of TC wind speed inducing a more powerful storm surge; and to a lesser extent due to sea level rise providing a higher baseline for the storm surge. We also show that the sensitivity of the results to the choice of TC intensification is approximately in the same range as for the tide assumption. We note that our attribution statements are, as commonly in the climate (impacts) attribution literature, purely statistical; that is, we do not make any claims about whether or to what extent any individual person may have been displaced because of climate change. Our methodology and results are subject to a variety of limitations and uncertainties, primarily related to the models (coastal, fluvial, DEM) and underlying datasets (population, displacement). Additional sources of uncertainty are the counterfactual input quantities (SLR, wind speed intensification), impact flood levels, and tide assumption, for which we perform sensitivity analyses."

Second last paragraph of the Discussion and conclusions:

Lines 720-726: "Framing the risk of tropical cyclones in the context of climate change in an event-specific rather than a probabilistic manner also allows us to assign absolute numbers of attributable displacements, which raises risk awareness in a more tangible way. Even though these numbers include substantial and important uncertainties related to the models, datasets and counterfactual assumptions, as discussed above, they provide an informative quantitative indication of the additional risk posed by climate change to communities affected by one of the worst natural disasters in recent history. "

Report #2 by Anonymous referee #2

Overview

This study aims to model and identify the 'excess' population displacement triggered by tropical cyclone Idai that can be attributed to climate change. By utilising a 'storyline approach' the study compares actual recorded displacement against estimated levels of displacement derived from counterfactual scenarios of mean sea level and maximum wind conditions without contributions of climate change. The conclusion of the study is that the impact of climate change has increased displacement risk by between 3.1 to 3.5%, corresponding to 16,000 – 17,000 displaced people.

The approach and results are interesting and of value in exploring the impact of climate change on environmental shocks and stressors, and the resulting affect this will have on patterns of human mobility.

The revisions that have taken place since the first review have significantly strengthened the paper. The assumptions on which the modelling approach is based are clearly outlined. The results of the study are much more clearly elucidated, alongside the limitations of results. This contextualises the author's use of a 'storyline approach' as opposed to, for example, a probabilistic approach and highlights the utility of this framing in terms of raising awareness of potential risks.

This is now a good paper and worthy of publication. This recommendation is made both in terms of the results themselves and the blueprint that it offers for future studies. As the authors highlight, the use of counterfactuals and 'storyline' approaches have potential in event-based displacement attribution.

A few minor textual changes and edits are suggested below, including an update of a hyperlink.

Astract

26 "corresponding to about 2.7 to 3.2%." Can these percentages be clarified? I presume they are of recorded / estimated actual displacees.

Yes, this is correct, the percentages relate to the number of observed displacements. We have revised the sentence:

Lines 24-27: "Our main estimates indicate that climate change has increased displacement risk from this event by approximately 12,600 - 14,900 additional displaced persons, corresponding to about 2.7 to 3.2% of the observed displacements."

1 Introduction

32 missing space 'TCs pose'

Done, thank you.

45 Suggest breaking sentence to improve clarity: '… very intense TCs (category 4-5 on the Saffir-Simpson scale). This is…'

We have revised the sentence as following:

Lines 47-51: "At the same time, global climate change is expected to alter TC characteristics, resulting in an increase in overall TC intensity (maximum wind speed and precipitation) and hence in the frequency of very intense TCs (category 4-5 on the Saffir-Simpson scale) (Knutson et al., 2020). Primarily, this is the result of an increase in potential intensity due to warmer sea surface temperatures (SST) (Emanuel, 2005, 2013, 1987)."

2 Methods

2.1 Counterfactuals

143 Mean sea level rise is expressed here in centimeters ('23cm') whereas later in the paragraph the IPCC rates are expressed in mm yr-1. Using millimeters in this instance (230mm, for example) is suggested.

Done, thank you.

Lines 144-145: "Total global mean sea level has risen by approximately 230 mm since the turn of the 20th century (Church and White, 2011)"

2.3 Inland Flood Depth Estimation

304 The location of the source code referenced has changed and the link should be updated to

https://github.com/NRCan/RICorDE/tree/main.

Indeed, the location has changed. We have replaced it with your suggestion. Thank you.

2.5 Displacements

342-347 This is a useful detail of the limitations of GIDD information

357-362 Again useful section which acknowledges the complexity of driver of displacement which helps place the results of this paper in context.

Thank you, we are glad to hear this.

398-399 & 401 There is an inconsistency between 'flood depth threshold' and 'flood-depth threshold', where the latter is preferred.

Thank you for pointing this out, we have changed the text and tables accordingly, and are now using only "flood-depth threshold".

4 Discussions and conclusions

545-549 This is a clear exposition of the findings.

562 Additional space 'ideally coupled'

Done, thank you.

674 onwards Useful framing of the 'storyline approach' in terms of its potential role in raising awareness of risks

697 Hyphen required in 'Idai-like'

The expression already contains a hyphen:

Lines 746-749: "Even though these increases may vary between basins, an enhanced displacement risk due to Idai-like TCs needs to be accounted for in the next decades, especially if future changes in exposure due to population growth and urbanization are considered."